# Subgroup-based Rank-1 Lattice Quasi-Monte Carlo

**Yueming Lyu**[*]
Australian Artificial Intelligence Institute
University of Technology Sydney
yueminglyu@gmail.com

**Yuan Yuan**
CSAIL
Massachusetts Institute of Technology
miayuan@mit.edu

**Ivor W. Tsang**[*]
Australian Artificial Intelligence Institute
University of Technology Sydney
Ivor.Tsang@uts.edu.au

## Abstract

Quasi-Monte Carlo (QMC) is an essential tool for integral approximation, Bayesian inference, and sampling for simulation in science, etc. In the QMC area, the rank-1 lattice is important due to its simple operation, and nice properties for point set construction. However, the construction of the generating vector of the rank-1 lattice is usually time-consuming because of an exhaustive computer search. To address this issue, we propose a simple closed-form rank-1 lattice construction method based on group theory. Our method reduces the number of distinct pairwise distance values to generate a more regular lattice. We theoretically prove a lower and an upper bound of the minimum pairwise distance of any non-degenerate rank-1 lattice. Empirically, our methods can generate a near-optimal rank-1 lattice compared with the Korobov exhaustive search regarding the $l_1$-norm and $l_2$-norm minimum distance. Moreover, experimental results show that our method achieves superior approximation performance on benchmark integration test problems and kernel approximation problems.

## 1 Introduction

Integral operation is critical in a large amount of interesting machine learning applications, e.g. kernel approximation with random feature maps [29], variational inference in Bayesian learning [3], generative modeling and variational autoencoders [15]. Directly calculating an integral is usually infeasible in these real applications. Instead, researchers usually try to find an approximation for the integral. A simple and conventional approximation is Monte Carlo (MC) sampling, in which the integral is approximated by calculating the average of the i.i.d. sampled integrand values. Monte Carlo (MC) methods [12] are widely studied with many techniques to reduce the approximation error, which includes importance sampling and variance reduction techniques and more [1].

To further reduce the approximation error, Quasi-Monte Carlo (QMC) methods utilize a low discrepancy point set instead of the i.i.d. sampled point set used in the standard Monte Carlo method. There are two main research lines in the area of QMC [8, 25], i.e., the digital nets/sequences and lattice rules. The Halton sequence and the Sobol sequence are the widely used representatives of digital sequences [8]. Compared with digital nets/sequences, the points set of lattice rules preserve the properties of lattice. The points partition the space into small repeating cells. Among previous research on the lattice rules, Korobov introduced integration lattice rules in [16] for an integral approximation of the periodic integrands. [33] proves that there also exist good lattice rules for

---

[*]Correspondence to: yueminglyu@gmail.com and Ivor.Tsang@uts.edu.au

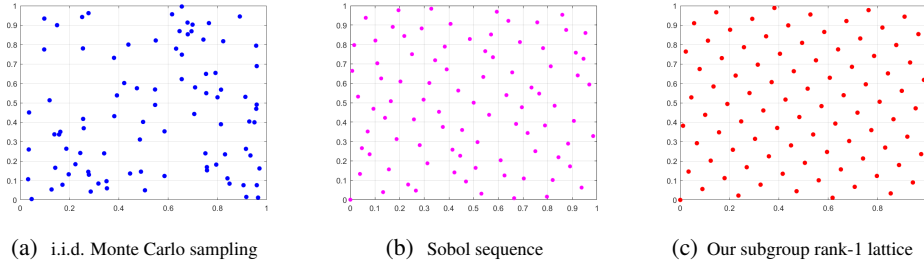

(a)  i.i.d. Monte Carlo sampling          (b)  Sobol sequence          (c)  Our subgroup rank-1 lattice

Figure 1: The 89 points constructed by i.i.d. Monte Carlo sampling, Sobol sequence and our subgroup rank-1 lattice on $[0, 1]^2$.

non-periodic integrands. According to general lattice rules, a point set is usually constructed by enumerating the integer vectors and multiplying them with an invertible generator matrix. A general lattice rule has to check each constructed point to see whether it is inside a unit cube and discard it if it is not. The process is repeated until we reach the desired number of points. This construction process is inefficient since the checking step is required for every point. Note that rescaling the unchecked points by the maximum norm of all the points may lead to non-uniform points set in the cube.

An interesting special case of the lattice rules is the rank-1 lattice, which only requires one generating vector to construct the whole point set. Given the generating vector, rank-1 lattices can be obtained by a very simple construction form. It is thus much more efficient to construct the point set with the simple construction form. Compared with the general lattice rule, the construction form of the rank-1 lattice has already guaranteed the constructed point to be inside the unit cube, therefore, no further checks are required. We refer to [8] and [25] for a more detailed survey of QMC and rank-1 lattice.

Although the rank-1 lattice can derive a simple construction form, obtaining the generating vector remains difficult. Most methods [17, 26, 9, 21, 20, 18, 27] in the literature rely on an exhaustive computer search by optimizing some criteria to find a good generating vector. Korobov [17] suggests searching the generating vector in a form of $[1, \alpha, \alpha^2, \cdots, \alpha^{d-1}]$ with $\alpha \in \{1, \cdots, n-1\}$, where $d$ is the dimension and $n$ is the number of points, such that the greatest common divisor of $\alpha$ and $n$ equals to 1. Sloan et al. study the component-by-component construction for the lattice rules [32]. It is a greedy search that is faster than an exhaustive search. Nuyens et al. [26] propose a fast algorithm to construct the generating vector using a component-by-component search method. Although the exhaustive checking steps are avoided compared with general lattice rules, the rank-1 lattice still requires a brute-force search for the generating vector, which is still very time-consuming, especially when the dimension and the number of points are large.

To address this issue, we propose a closed-form rank-1 lattice rule that directly computes a generating vector without any search process. To generate a more evenly spaced lattice, we propose to reduce the number of distinct pairwise distance in the lattice point set to make the lattice more regular w.r.t. the minimum toroidal distance [11]. Larger minimum toroidal distance means more regular. Based on group theory, we derive that if the generating vector $\boldsymbol{z}$ satisfies the condition that set $\{\boldsymbol{z}, -\boldsymbol{z}\} := \{z_1, \cdots, z_d, -z_1, \cdots, -z_d\}$ is a subgroup of the multiplicative group of integers modulo $n$, where $n$ is the number of points, then the number of distinct pairwise distance can be efficiently reduced. We construct the generating vector by ensuring this condition. With the proposed subgroup-based rank-1 lattice, we can construct a more evenly spaced lattice. An illustration of the generated lattice is shown in Figure 1. Our contributions are summarized as follows:

- We propose a simple and efficient closed-form method for rank-1 lattice construction, which does not require the time-consuming exhaustive computer search that previous rank-1 lattice algorithms rely on. A side product is a closed-form method to generate QMC points set on sphere $\mathbb{S}^{d-1}$ with bounded mutual coherence, which is presented in Appendix.

- We generate a more regular lattice by reducing the number of distinct pairwise distances. We prove a lower and an upper bound for the minimum $l_1$-norm-based and $l_2$-norm-based toroidal distance of the rank-1 lattice. Theoretically, our constructed lattice is the optimal rank-1 lattice for maximizing the minimum toroidal distance when the number of points $n$ is a prime number and $n = 2d + 1$.

- Empirically, the proposed method generates near-optimal rank-1 lattice compared with the Korobov search method in maximizing the minimum of the $l_1$-norm-based and $l_2$-norm-based toroidal distance.
- Our method obtains better approximation accuracy on benchmark test problems and kernel approximation problem.

## 2 Background

We first give the definition and the properties of lattices in Section 2.1. Then we introduce the minimum distance criterion for lattice construction in Section 2.2.

### 2.1 The Lattice

A $d$-dimensional lattice $\Lambda$ is a set of points that contains no limit points and satisfies [22]

$$\forall \boldsymbol{x}, \boldsymbol{x}' \in \Lambda \Rightarrow \boldsymbol{x} + \boldsymbol{x}' \in \Lambda \text{ and } \boldsymbol{x} - \boldsymbol{x}' \in \Lambda. \tag{1}$$

A widely known lattice is the unit lattice $\mathbb{Z}^d$ whose components are all integers. A general lattice is constructed by a generator matrix. Given a generator matrix $\boldsymbol{B} \in \mathbb{R}^{d \times d}$, a $d$-dimensional lattice $\Lambda$ can be constructed as

$$\Lambda = \{\boldsymbol{B}\boldsymbol{y} \big| \, \boldsymbol{y} \in \mathbb{Z}^d\}. \tag{2}$$

A generator matrix is not unique to a lattice $\Lambda$, namely, a lattice $\Lambda$ can be obtained from a different generator matrices.

A lattice point set for integration is constructed as $\Lambda \cap [0, 1)^d$. This step may require an additional search (or check) for all the points inside the unit cube.

A rank-1 lattice is a special case of the general lattice, which has a simple operation for point set construction instead of directly using Eq.(2). A rank-1 lattice point set can be constructed as

$$\boldsymbol{x}_i := \left\langle \frac{i\boldsymbol{z}}{n} \right\rangle, i \in \{0, 1, \cdots, n - 1\}, \tag{3}$$

where $\boldsymbol{z} \in \mathbb{Z}^d$ is the so-called generating vector, and the big $\langle \cdot \rangle$ denotes the operation of taking the fractional part of the input number elementwise. Compared with the general lattice rule, the construction form of the rank-1 lattice already ensures the constructed points to be inside the unit cube without the need for any further checks.

Given a rank-1 lattice set $X$ in the unit cube, we can also construct a randomized point set. Sample a random variable $\boldsymbol{\Delta} \sim Uniform[0, 1]^d$, we can construct a point set $\widetilde{X}$ by random shift as [8]

$$\widetilde{X} = \langle X + \boldsymbol{\Delta} \rangle. \tag{4}$$

### 2.2 The separating distance of a lattice

Several criteria have been studied in the literature for good lattice construction through computer search. Worst case error is one of the most widely used criteria for functions in a reproducing kernel Hilbert space (RKHS) [8]. However, this criterion requires the prior knowledge of functions and the assumption of the RKHS. Without assumptions of the functions, it is reasonable to construct a good lattice by designing an evenly spaced point set. Minimizing the covering radius is a good way for evenly spaced point set construction.

As minimizing the covering radius of the lattice is equivalent to maximizing the packing radius [7], we can construct the point set through maximizing the packing radius (separating distance) of the lattice. Define the covering radius and packing radius of a set of points $X = \{x_1, ..., x_N\}$ as Eq.(5) and Eq.(6), respectively:

$$h_X = \sup_{x \in \mathcal{X}} \min_{x_k \in X} \|x - x_k\|, \tag{5}$$

$$\rho_X = \frac{1}{2} \min_{\substack{x_i, x_j \in X, \\ x_i \neq x_j}} \|x_i - x_j\|. \tag{6}$$

The $l_p$-norm-based toroidal distance [11] between two lattice points $\mathbf{x} \in [0,1]^d$ and $\mathbf{x}' \in [0,1]^d$ can be defined as:

$$\|\mathbf{x} - \mathbf{x}'\|_{T_p} := \left( \sum_{i=1}^{d} (\min(|x_i - x_i'|, 1 - |x_i - x_i'|))^p \right)^{\frac{1}{p}} \tag{7}$$

Because the difference (subtraction) between two lattice points is still a lattice point, and a rank-1 lattice has a period 1, the packing radius $\rho_X$ of a rank-1 lattice can be calculated as

$$\rho_X = \min_{\mathbf{x} \in X \setminus \mathbf{0}} \frac{1}{2} \|\mathbf{x}\|_{T_2}, \tag{8}$$

where $\|\mathbf{x}\|_{T_2}$ denotes the $l_2$-norm-based toroidal distance between $\mathbf{x}$ and $\mathbf{0}$, symbol $X \setminus \mathbf{0}$ denotes the set $X$ excludes the point $\mathbf{0}$. This formulation calculates the packing radius with a time complexity of $\mathcal{O}(Nd)$ rather than $\mathcal{O}(N^2 d)$ in pairwise comparison. However, the computation of the packing radius is not easily accelerated by fast Fourier transform due to the minimum operation in Eq.(8).

## 3 Subgroup-based Rank-1 Lattice

In this section, we derive our construction of a rank-1 lattice based on the subgroup theory. Then we analyze the properties of our method. We provide detailed proofs in the supplement.

### 3.1 Construction of the Generating Vector

From the definition of rank-1 lattice, we know the packing radius of rank-1 lattice with $n$ points can be reformulated as

$$\rho_X = \min_{i \in \{1, \cdots, n-1\}} \frac{1}{2} \|\mathbf{x}_i\|_{T_2}, \tag{9}$$

where

$$\boldsymbol{x}_i := \frac{i\boldsymbol{z} \bmod n}{n}, i \in \{1, ..., n-1\}. \tag{10}$$

Then, we have

$$\begin{aligned} \rho_X &= \min_{i \in \{1, \cdots, n-1\}} \frac{1}{2} \left\| \min\left( \frac{i\boldsymbol{z} \bmod n}{n}, \frac{n - i\boldsymbol{z} \bmod n}{n} \right) \right\|_2 \\ &= \min_{i \in \{1, \cdots, n-1\}} \frac{1}{2} \left\| \min\left( \frac{i\boldsymbol{z} \bmod n}{n}, \frac{(-i\boldsymbol{z}) \bmod n}{n} \right) \right\|_2, \end{aligned} \tag{11}$$

where $\min(\cdot, \cdot)$ denotes the elementwise min operation between two inputs.

Suppose $n$ is a prime number, from number theory, we know that for a primitive root $g$, the residue of $\{g^0, g^1, \cdots, g^{n-2}\}$ modulo $n$ forms a cyclic group under multiplication, and $g^{n-1} \equiv 1 \bmod n$. Since $(g^{\frac{n-1}{2}})^2 = g^{n-1} \equiv 1 \bmod n$, we know that $g^{\frac{n-1}{2}} \equiv -1 \bmod n$.

Because of the one-to-one correspondence between the residue of $\{g^0, g^1, \cdots, g^{n-2}\}$ modulo $n$ and the set $\{1, 2, \cdots, n-1\}$, we can construct the generating vector as

$$\boldsymbol{z} = [g^{m_1}, g^{m_2}, \cdots, g^{m_d}] \bmod n \tag{12}$$

without loss of generality, where $m_1, \cdots, m_d$ are integer components to be designed. Denote $\bar{\boldsymbol{z}} = [g^{\frac{n-1}{2} + m_1}, g^{\frac{n-1}{2} + m_2}, \cdots, g^{\frac{n-1}{2} + m_d}] \bmod n$, maximizing the separating distance $\rho_X$ is equivalent to maximizing

$$J = \min_{k \in \{0, \cdots, n-2\}} \left\| \min(g^k \boldsymbol{z} \bmod n, g^k \bar{\boldsymbol{z}} \bmod n) \right\|_2. \tag{13}$$

Suppose $2d$ divides $n-1$, i.e., $2d | (n-1)$, by setting $m_i = g^{\frac{(i-1)(n-1)}{2d}}$ for $i \in \{1, \cdots, d\}$, we know that $H = \{g^{m_1}, g^{m_2}, \cdots, g^{m_d}, g^{\frac{n-1}{2} + m_1}, g^{\frac{n-1}{2} + m_2}, \cdots, g^{\frac{n-1}{2} + m_d}\}$ is equivalent to setting $\{g^0, g^{\frac{n-1}{2d}}, \cdots, g^{\frac{(2d-1)(n-1)}{2d}}\} \bmod n$, and it forms a subgroup of the group $\{g^0, g^1, \cdots, g^{n-2}\} \bmod n$. From Lagrange's theorem in group theory [10], we know that the cosets of the subgroup $H$ partition

the entire group $\{g^0, g^1, \cdots, g^{n-2}\}$ into equal-size, non-overlapping sets, and the number of cosets of $H$ is $\frac{n-1}{2d}$. Thus, we know that distance $\min(g^k z \bmod n, g^k \bar{z} \bmod n)$ for $k \in \{0, \cdots, n-2\}$ has $\frac{n-1}{2d}$ different values, and there are the same numbers of items for each value.

Thus, we can construct the generating vector as

$$z = [g^0, g^{\frac{n-1}{2d}}, g^{\frac{2(n-1)}{2d}}, \cdots, g^{\frac{(d-1)(n-1)}{2d}}] \bmod n. \tag{14}$$

In this way, the constructed rank-1 lattice is more regular as it has few different distinct pairwise distance values, and for each distance, the same number of items obtain this value. Usually, the constructed regular lattice is more evenly spaced, and it has a large minimum pairwise distance. We confirm this empirically through extensive experiments in Section 5.

We summarize our construction method and the properties of the constructed rank-1 lattice in Theorem 1.

**Theorem 1.** *Suppose $n$ is a prime number and $2d|(n-1)$. Let $g$ be a primitive root of $n$. Let $z = [g^0, g^{\frac{n-1}{2d}}, g^{\frac{2(n-1)}{2d}}, \cdots, g^{\frac{(d-1)(n-1)}{2d}}] \bmod n$. Construct a rank-1 lattice $X = \{x_0, \cdots, x_{n-1}\}$ with $x_i = \frac{iz \bmod n}{n}, i \in \{0, ..., n-1\}$. Then, there are $\frac{n-1}{2d}$ distinct pairwise toroidal distance values among $X$, and each distance value is taken by the same number of pairs in $X$.*

As shown in Theorem 1, our method can construct regular rank-1 lattice through a very simple closed-form construction, which does not require any exhaustive computer search.

## 3.2 Regular Property of Rank-1 Lattice

We show the regular property of rank-1 lattices in terms of $l_p$-norm-based toroidal distance.

**Theorem 2.** *Suppose $n$ is a prime number and $n \geq 2d+1$. Let $z = [z_1, z_2, \cdots, z_d]$ with $1 \leq z_k \leq n-1$. Construct a rank-1 lattice $X = \{x_0, \cdots, x_{n-1}\}$ with $x_i = \frac{iz \bmod n}{n}, i \in \{0, ..., n-1\}$ and $z_i \neq z_j$. Then, the minimum pairwise toroidal distance can be bounded as*

$$\frac{d(d+1)}{2n} \leq \min_{i,j \in \{0, \cdots, n-1\}, i \neq j} \|\mathbf{x}_i - \mathbf{x}_j\|_{T_1} \leq \frac{(n+1)d}{4n} \tag{15}$$

$$\frac{\sqrt{6d(d+1)(2d+1)}}{6n} \leq \min_{i,j \in \{0, \cdots, n-1\}, i \neq j} \|\mathbf{x}_i - \mathbf{x}_j\|_{T_2} \leq \sqrt{\frac{(n+1)d}{12n}}, \tag{16}$$

*where $\|\cdot\|_{T_1}$ and $\|\cdot\|_{T_2}$ denote the $l_1$-norm-based toroidal distance and the $l_2$-norm-based toroidal distance, respectively.*

Theorem 2 gives the upper and lower bounds of the minimum pairwise distance of any non-degenerate rank-1 lattice. The term 'non-degenerate' means that the elements in the generating vector are not equal, i.e., $z_i \neq z_j$.

We now show that our subgroup-based rank-1 lattice can achieve the optimal minimum pairwise distance when $n = 2d+1$ is a prime number.

**Corollary 1.** *Suppose $n = 2d+1$ is a prime number. Let $g$ be a primitive root of $n$. Let $z = [g^0, g^{\frac{n-1}{2d}}, g^{\frac{2(n-1)}{2d}}, \cdots, g^{\frac{(d-1)(n-1)}{2d}}] \bmod n$. Construct rank-1 lattice $X = \{x_0, \cdots, x_{n-1}\}$ with $x_i = \frac{iz \bmod n}{n}, i \in \{0, ..., n-1\}$. Then, the pairwise toroidal distance of the lattice $X$ attains the upper bound.*

$$\|\mathbf{x}_i - \mathbf{x}_j\|_{T_1} = \frac{(n+1)d}{4n}, \forall i,j \in \{0, \cdots, n-1\}, i \neq j, \tag{17}$$

$$\|\mathbf{x}_i - \mathbf{x}_j\|_{T_2} = \sqrt{\frac{(n+1)d}{12n}}, \forall i,j \in \{0, \cdots, n-1\}, i \neq j. \tag{18}$$

Corollary 1 shows a case when our subgroup rank-1 lattice obtains the maximum minimum pairwise toroidal distance. It is useful for expensive black-box functions, where the number of function queries is small. Empirically, we find that our subgroup rank-1 lattice can achieve near-optimal pairwise toroidal distance in many other cases.

Table 1: Minimum $l_1$-norm-based toroidal distance of rank-1 lattice constructed by different methods.

| d=50 | | n=101 | 401 | 601 | 701 | 1201 | 1301 | 1601 | 1801 | 1901 | 2801 |
|---|---|---|---|---|---|---|---|---|---|---|---|
| | SubGroup | **12.624** | **11.419** | **11.371** | **11.354** | **11.029** | **10.988** | 10.541 | 10.501 | 10.454 | **10.748** |
| | Hua [13] | 10.426 | 10.421 | 9.8120 | 10.267 | 10.074 | 9.3982 | 9.5890 | 9.5175 | 8.9868 | 9.2260 |
| | Korobov [17] | **12.624** | **11.419** | **11.371** | **11.354** | **11.029** | **10.988** | 10.665 | 10.561 | 10.701 | **10.748** |
| d=100 | | 401 | 601 | 1201 | 1601 | 1801 | 2801 | 3001 | 4001 | 4201 | 4801 |
| | SubGroup | **24.097** | **23.760** | 22.887 | **23.342** | 22.711 | **23.324** | 22.233 | **22.437** | 22.573 | 21.190 |
| | Hua [13] | 21.050 | 21.251 | 21.205 | 20.675 | 19.857 | 20.683 | 20.700 | 19.920 | 19.967 | 20.574 |
| | Korobov [17] | **24.097** | **23.760** | 23.167 | **23.342** | 22.893 | **23.324** | 22.464 | **22.437** | 22.573 | 22.188 |
| d=200 | | 401 | 1201 | 1601 | 2801 | 4001 | 4801 | 9601 | 12401 | 14401 | 15601 |
| | SubGroup | **50.125** | **48.712** | 47.500 | 47.075 | **47.810** | 45.957 | 45.819 | **46.223** | 43.982 | **45.936** |
| | Hua [13] | 43.062 | 43.057 | 43.052 | 43.055 | 43.053 | 43.055 | 43.053 | 42.589 | 42.558 | 42.312 |
| | Korobov [17] | **50.125** | **48.712** | 47.660 | 47.246 | **47.810** | 46.686 | 46.154 | **46.223** | 45.949 | **45.936** |
| d=500 | | 3001 | 4001 | 7001 | 9001 | 13001 | 16001 | 19001 | 21001 | 24001 | 28001 |
| | SubGroup | **121.90** | **121.99** | 119.60 | 118.63 | **120.23** | **119.97** | 116.41 | **120.56** | **120.24** | 113.96 |
| | Hua [13] | 108.33 | 108.33 | 108.33 | 108.33 | 108.33 | 108.33 | 108.33 | 108.33 | 108.33 | 108.33 |
| | Korobov [17] | **121.90** | **121.99** | 120.46 | 120.16 | **120.23** | **119.97** | 119.41 | **120.56** | **120.24** | 118.86 |

# 4 QMC for Kernel Approximation

Another application of our subgroup rank-1 lattice is kernel approximation. Kernel approximation has been widely studied. A random feature maps is a promising way for kernel approximation. Rahimi et al. study the shift-invariant kernels by Monte Carlo sampling [29]. Yang et al. suggest employing QMC for kernel approximation [35, 2]. Several previous methods work on the construction of structured feature maps for kernel approximation [19, 6, 23]. Apart from other kernel approximation methods designed for specific kernels, QMC can serve as a plug-in for any integral representation of kernels to improve kernel approximation. We include this section to be self-contained.

From Bochner's Theorem, shift invariant kernels can be expressed as an integral [29]

$$\mathrm{K}(\boldsymbol{x}, \boldsymbol{y}) = \int_{\mathbb{R}^d} e^{-i(\boldsymbol{x}-\boldsymbol{y})^\top \mathbf{w}} p(\mathbf{w}) \mathbf{dw}, \tag{19}$$

where $i = \sqrt{-1}$, and $p(\mathbf{w})$ is a probability density. $p(\mathbf{w}) = p(-\mathbf{w}) \geq 0$ ensure the imaginary parts of the integral vanish. Eq.(19) can be rewritten as

$$\mathrm{K}(\boldsymbol{x}, \boldsymbol{y}) = \int_{[0,1]^d} e^{-i(\boldsymbol{x}-\boldsymbol{y})^\top \Phi^{-1}(\boldsymbol{\epsilon})} d\boldsymbol{\epsilon}. \tag{20}$$

We can approximate the integral Eq.(19) by using our subgroup rank-1 lattice according to the QMC approximation in [35, 34]

$$\mathrm{K}(\boldsymbol{x}, \boldsymbol{y}) = \int_{[0,1]^d} e^{-i(\boldsymbol{x}-\boldsymbol{y})^\top \Phi^{-1}(\boldsymbol{\epsilon})} d\boldsymbol{\epsilon} \approx \frac{1}{n} \sum_{i=1}^{n} e^{-i(\boldsymbol{x}-\boldsymbol{y})^\top \Phi^{-1}(\boldsymbol{\epsilon}_i)} = \langle \Psi(\boldsymbol{x}), \Psi(\boldsymbol{y}) \rangle, \tag{21}$$

where $\Psi(\boldsymbol{x}) = \frac{1}{\sqrt{n}} \left[ e^{-i\boldsymbol{x}^\top \Phi^{-1}(\boldsymbol{\epsilon}_1)}, \cdots, e^{-i\boldsymbol{x}^\top \Phi^{-1}(\boldsymbol{\epsilon}_n)} \right]$ is the feature map of the input $\boldsymbol{x}$.

# 5 Experiments

In this section, we first evaluate the minimum distance generated by our subgroup rank-1 lattice in section 5.1. We then evaluate the subgroup rank-1 lattice on integral approximation tasks and kernel approximation task in section 5.2 and 5.3, respectively.

## 5.1 Evaluation of the minimum distance

We evaluate the minimum distance of our subgroup rank-1 lattice by comparing with Hua's method [13] and the Korobov [17] searching method. We denote 'SubGroup' as our subgroup rank-1 lattice, 'Hua' as rank-1 lattice constructed by Hua's method [13], and 'Korobov' as rank-1 lattice constructed by exhaustive computer search in Korobov form [17].

We set the dimension $d$ as in $\{50, 100, 200, 500\}$. For each dimension $d$, we set the number of points $n$ as the first ten prime numbers such that $2d$ divides $n-1$, i.e., $2d \big| (n-1)$. The minimum $l_1$-norm-based toroidal distance and the minimum $l_2$-norm-based toroidal distance for each dimension are reported in Table 5.1 and Table 2, respectively. The larger the distance, the better.

Table 2: Minimum $l_2$-norm-based toroidal distance of rank-1 lattice constructed by different methods.

| | | n=101 | 401 | 601 | 701 | 1201 | 1301 | 1601 | 1801 | 1901 | 2801 |
|---|---|---|---|---|---|---|---|---|---|---|---|
| d=50 | SubGroup | **2.0513** | **1.9075** | **1.9469** | **1.9196** | **1.8754** | 1.8019 | 1.8008 | **1.8709** | 1.7844 | 1.7603 |
| | Hua [13] | 1.7862 | 1.7512 | 1.7293 | 1.7049 | 1.7326 | 1.6295 | 1.6659 | 1.6040 | 1.5629 | 1.5990 |
| | Korobov [17] | **2.0513** | **1.9075** | **1.9469** | **1.9196** | **1.8754** | 1.8390 | 1.8356 | **1.8709** | 1.8171 | 1.8327 |
| | | 401 | 601 | 1201 | 1601 | 1801 | 2801 | 3001 | 4001 | 4201 | 4801 |
| d=100 | SubGroup | **2.8342** | **2.8143** | 2.7077 | **2.7645** | **2.7514** | 2.6497 | 2.6337 | 2.6410 | 2.6195 | 2.5678 |
| | Hua [13] | 2.5385 | 2.5739 | 2.4965 | 2.4783 | 2.4132 | 2.5019 | 2.4720 | 2.4138 | 2.4537 | 2.4937 |
| | Korobov [17] | **2.8342** | **2.8143** | 2.7409 | **2.7645** | **2.7514** | 2.6956 | 2.6709 | 2.6562 | 2.6667 | 2.6858 |
| | | 401 | 1201 | 1601 | 2801 | 4001 | 4801 | 9601 | 12401 | 14401 | 15601 |
| d=200 | SubGroup | **4.0876** | **3.9717** | **3.9791** | 3.8425 | **3.9276** | 3.8035 | 3.7822 | **3.8687** | 3.6952 | 3.8370 |
| | Hua [13] | 3.7332 | 3.7025 | 3.6902 | 3.6944 | 3.7148 | 3.6936 | 3.6571 | 3.5625 | 3.6259 | 3.5996 |
| | Korobov [17] | **4.0876** | **3.9717** | **3.9791** | 3.9281 | **3.9276** | 3.9074 | 3.8561 | **3.8687** | 3.8388 | 3.8405 |
| | | 3001 | 4001 | 7001 | 9001 | 13001 | 16001 | 19001 | 21001 | 24001 | 28001 |
| d=500 | SubGroup | **6.3359** | **6.3769** | 6.3141 | 6.2131 | **6.2848** | 6.2535 | 6.0656 | **6.2386** | **6.2673** | 6.1632 |
| | Hua [13] | 5.9216 | 5.9216 | 5.9215 | 5.9215 | 5.9216 | 5.9216 | 5.9215 | 5.9215 | 5.8853 | 5.9038 |
| | Korobov [17] | **6.3359** | **6.3769** | 6.3146 | 6.2960 | **6.2848** | 6.2549 | 6.2611 | **6.2386** | **6.2673** | 6.2422 |

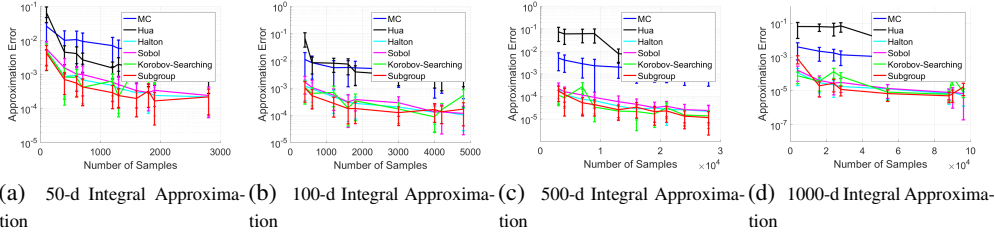

(a) 50-d Integral Approxima- (b) 100-d Integral Approxima- (c) 500-d Integral Approxima- (d) 1000-d Integral Approxima-
tion tion tion tion

Figure 2: Mean approximation error over 50 independent runs.error bars are with in $1\times$ std.

We can observe that our subgroup rank-1 lattice achieves consistently better (larger) minimum distances than Hua's method in all the cases. Moreover, we see that subgroup rank-1 lattice obtains, in 20 out of 40 cases, the same $l_2$-norm-based toroidal distance and in 24 out of 40 cases the same $l_1$-norm-based toroidal distance compared with the exhaustive computer search in Korobov form. The experiments show that our subgroup rank-1 lattice achieves the optimal toroidal distance in exhaustive computer searches in Korobov form in over half of all the cases. Furthermore, the experimental result shows that our subgroup rank-1 lattice obtains a competitive distance compared with the exhaustive Korobov search in the remaining cases. Note that our subgroup rank-1 lattice is a closed-form construction which does not require computer search, making our method more appealing and simple to use.

**Time Comparison of Korobov searching and our sub-group rank-1 lattice.** The table below shows the time cost (seconds) for lattice construction. The run time for Korobov searching grows fast to hours. Our method can run in less than one second, achieving a $10^4\times$ to $10^5\times$ speed-up. The speed-up increases when $n$ and $d$ becomes larger.

| | | n=3001 | 4001 | 7001 | 9001 | 13001 | 16001 | 19001 | 21001 | 24001 | 28001 |
|---|---|---|---|---|---|---|---|---|---|---|---|
| d=500 | SubGroup | 0.0185 | 0.0140 | 0.0289 | 0.043 | 0.0386 | 0.0320 | 0.0431 | 0.0548 | 0.0562 | 0.0593 |
| | Korobov | 34.668 | 98.876 | 152.86 | 310.13 | 624.56 | 933.54 | 1308.9 | 1588.5 | 2058.5 | 2815.9 |
| | | n=4001 | 16001 | 24001 | 28001 | 54001 | 70001 | 76001 | 88001 | 90001 | 96001 |
| d=1000 | SubGroup | 0.0388 | 0.0618 | 0.1041 | 0.1289 | 0.2158 | 0.2923 | 0.3521 | 0.4099 | 0.5352 | 0.5663 |
| | Korobov | 112.18 | 1849.4 | 4115.9 | 5754.6 | 20257 | 34842 | 43457 | 56798 | 56644 | 69323 |

## 5.2 Integral approximation

We evaluate our subgroup rank-1 lattice on the integration test problem

$$f(\boldsymbol{x}) := \exp\left(c\sum_{j=1}^{d} x_j j^{-b}\right) \tag{22}$$

$$I(f) := \int_{[0,1]^d} f(\boldsymbol{x})d\boldsymbol{x} = \prod_{j=1}^{d} \frac{\exp(cj^{-b}) - 1}{cj^{-b}}. \tag{23}$$

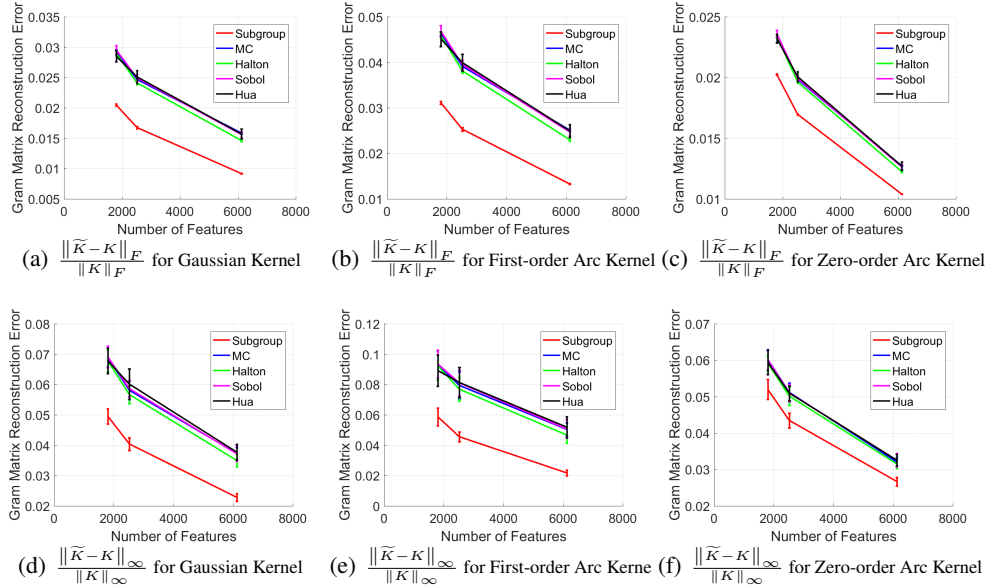

Figure 3: Relative Mean and Max Reconstruction Error for Gaussian, Zero-order and First-order Arc-cosine Kernel on DNA dataset. Error bars are within $1\times$ std.

We compare with i.i.d. Monte Carlo, a Hua's rank-1 lattice [13], Korobov searching rank-1 lattice [16], Halton sequence, and Sobol sequence [8]. For both Halton sequence and Sobol sequence, we use the scrambling technique suggested in [8]. For all the QMC methods, we use the random shift technique as in Eq.(4).

We fix $b = 2$ and $c = 1$ in all the experiments. We set dimension $d = 100$ and $d = 500$, respectively. We set the number of points $n$ as the first ten prime numbers such that $2d$ divides $n-1$, i.e., $2d\big|(n-1)$.

The mean approximation error ($\frac{|Q(f)-I(f)|}{|I(f)|}$) with error bars over 50 independent runs for each dimension $d$ is presented in Figure 2. We can see that Hua's method obtains a smaller error than i.i.d Monte Carlo on the 50-d problem, however, it becomes worse than MC on 500-d and 1000-d problems. Moreover, our subgroup rank-1 lattice obtains a consistent smaller error on all the tested problems than Hua and MC. In addition, our subgroup rank-1 lattice achieves a slightly better performance than Halton, Sobol and Korobov searching method.

## 5.3 Kernel approximation

We evaluate the performance of subgroup rank-1 lattice on kernel approximation tasks by comparing with other QMC baseline methods. We test the kernel approximation of the Gaussian kernel, the zeroth-order arc-cosine kernel, and the first-order arc-cosine kernel as in [6].

We compare subgroup rank-1 lattice with a Hua's rank-1 lattice [13], Halton sequence, Sobol sequence [8] and standard i.i.d. Monte Carlo sampling. For both the Halton sequence and Sobol sequence, we use the scrambling technique suggested in [8]. For both subgroup rank-1 lattice and Hua's rank-1 lattice, we use the random shift as in Eq.(4). We evaluate the methods on the DNA [28] and the SIFT1M [14] dataset over 50 independent runs. Each run contains 2000 random samples to construct the Gram matrix. The bandwidth parameter of Gaussian kernel is set to 15 in all the experiments.

The mean Frobenius norm approximation error ($\|\widetilde{K}-K\|_F/\|K\|_F$) and maximum norm approximation error ($\|\widetilde{K}-K\|_\infty/\|K\|_\infty$) with error bars on DNA [28] dataset are plotted in Figure 3. The results on SIFT1M [14] is given in Figure 6 in the supplement. The experimental result shows that subgroup rank-1 lattice consistently obtains a smaller approximation error compared with other baselines.

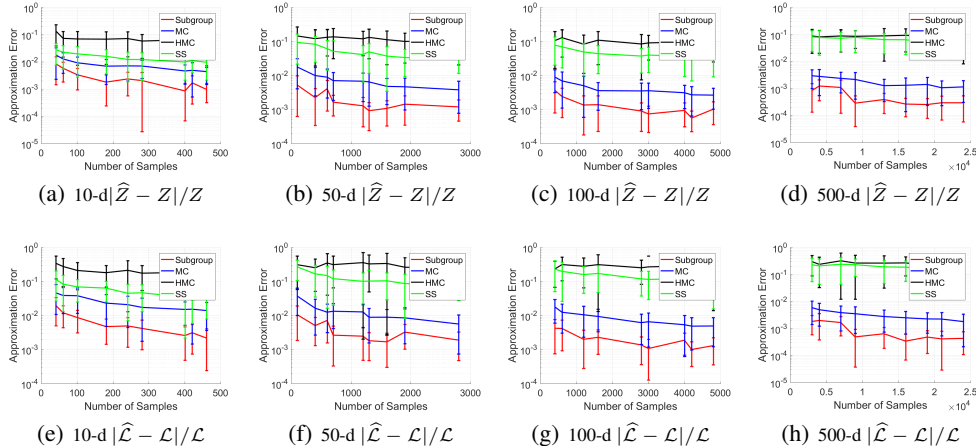

Figure 4: Mean approximation error over 50 independent runs. Error bars are with in $1\times$ std

## 5.4 Approximation on Graphical Model

For general Boltzmann machines with continuous state in $[0,1]$, the energy function of $\boldsymbol{x} \in [0,1]^d$ is defined as $E(\boldsymbol{x}) = -(\boldsymbol{x}^\top \boldsymbol{W} \boldsymbol{x} + \boldsymbol{b}^\top \boldsymbol{x})/d$. The normalization constant is $Z = \int_{[0,1]^d} \exp(-E(\boldsymbol{x}))d\boldsymbol{x}$. For inference, the marginal likelihood of observation $\boldsymbol{v} \in \mathbb{R}^d$ is $\mathcal{L}(\boldsymbol{v}) = \int_{[0,1]^d} \exp(-f(\boldsymbol{v})) \exp(-E(\boldsymbol{h}))/Z d\boldsymbol{h}$ with function $f(\boldsymbol{v}) = -(\boldsymbol{v}^\top \boldsymbol{W}_v \boldsymbol{v} + 2\boldsymbol{v}^\top \boldsymbol{W}_h \boldsymbol{h} + \boldsymbol{b}_v^\top \boldsymbol{v})/d$, where $\boldsymbol{h} \in \mathbb{R}^d$ denotes the hidden states.

We evaluate our method on approximation of the normalization constant and inference by comparing with i.i.d. Monte Carlo (MC), slice sampling (SS) and Hamiltonian Monte Carlo (HMC). We generate the elements of $\boldsymbol{W}$, $\boldsymbol{W}_v$, $\boldsymbol{W}_h$, $\boldsymbol{b}$ and $\boldsymbol{b}_v$ by sampling from standard Gaussian $\mathcal{N}(0,1)$. These parameters are fixed and kept the same for all the methods in comparison. For inference, we generate an observation $\boldsymbol{v} \in [0,1]^d$ by uniformly sampling and keep it fixed and same for all the methods. For SS and HMC, we use the *slicesample* function and *hmcSampler* function in MATLAB, respectively. We use the approximation of i.i.d. MC with $10^7$ samples as the pseudo ground-truth. The approximation errors $|\widehat{Z} - Z|/Z$ and $|\widehat{\mathcal{L}} - \mathcal{L}|/\mathcal{L}$ are shown in Fig.4(a)-4(d) and Fig.4(e)-4(h), respectively. our method consistently outperforms MC, HMC and SS on all cases. Moreover, our method is much cheaper than SS and HMC.

**Comparison to sequential Monte Carlo.** When the positive density region takes a large fraction of the entire domain, our method is very competitive. When it is only inside a small part of a large domain, our method may not be better than sequential adaptive sampling. In this case, it is interesting to take advantage of both lattice and adaptive sampling. E.g., one can employ our subgroup rank-1 lattice as a rough partition of the domain to find high mass regions, then take sequential adaptive sampling on the promising regions with the lattice points as the start points. Also, it is interesting to consider recursively apply our subgroup rank-1 lattice to refine the partition. Moreover, our subgroup-based rank-1 lattice enables black-box evaluation without the need for gradient information. In contrast, several sequential sampling methods, e.g., HMC, need a gradient of density function for sampling.

## 6 Conclusion

We propose a closed-form method for rank-1 lattice construction, which is simple and efficient without exhaustive computer search. Theoretically, we prove that our subgroup rank-1 lattice has few different pairwise distance values, which is more regular to be evenly spaced. Moreover, we prove a lower and an upper bound for the minimum toroidal distance of a non-degenerate rank-1 lattice. Empirically, our subgroup rank-1 lattice obtains near-optimal minimum toroidal distance compared with Korobov exhaustive search. Moreover, subgroup rank-1 lattice achieves smaller integration approximation error. In addition, we propose a closed-form method to generate QMC points set on sphere $\mathbb{S}^{d-1}$. We proved upper bounds of the mutual coherence of the generated points. Further, we show an example of CycleGAN training in the supplement. Our subgroup rank-1 lattice sampling and QMC on sphere can serve as an alternative for training generative models.

## Broader Impact

In this paper, we proposed a closed-form rank-one lattice construction based on group theory for Quasi-Monte Carlo. Our method does not require the time-consuming exhaustive computer search. Our method is a fundamental tool for integral approximation and sampling.

Our method may serve as a potential advance in QMC, which may have an impact on a wide range of applications that rely on integral approximation. It includes kernel approximation with feature map, variational inference in Bayesian learning, generative modeling, and variational autoencoders. This may bring useful applications and be beneficial to society and the community. Since our method focuses more on the theoretical side, the direct negative influences and ethical issues are negligible.

## Acknowledgement and Funding Disclosure

We thank the reviewers for their valuable comments and suggestions. Yueming Lyu was supported by UTS President Scholarship. Prof. Ivor W. Tsang was supported by ARC DP180100106 and DP200101328.

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
