[Supplementary Material]

**Organization:** In the supplement, we present the detailed proof of the Theorem 1, Theorem 2 and Corollary 1 in section A, section B and section C, respectively. We then present a subgroup-based QMC on sphere $\mathbb{S}^{d-1}$ in Section D. We give the detailed proof of Theorem 3 and Theorem 4 in section E and section F, respectively. We then present QMC for generative CycleGAN in section G and section H. At last, we present the experimental results of kernel approximation on SIFT1M dataset in Figure 7.

# A   Proof of Theorem 1

**Theorem.** *Suppose $n$ is a prime number and $2d|(n-1)$. Let $g$ be a primitive root of $n$. Let $\boldsymbol{z} = [g^0, g^{\frac{n-1}{2d}}, g^{\frac{2(n-1)}{2d}}, \cdots, g^{\frac{(d-1)(n-1)}{2d}}] \bmod n$. Construct a rank-1 lattice $X = \{\boldsymbol{x}_0, \cdots, \boldsymbol{x}_{n-1}\}$ with $\boldsymbol{x}_i = \frac{i\boldsymbol{z} \bmod n}{n}, i \in \{0, ..., n-1\}$. Then, there are $\frac{n-1}{2d}$ distinct pairwise toroidal distance values among $X$, and for each distance value, there are the same number of pairs that obtain this value.*

*Proof.* From the definition of the rank-1 lattice, we know that

$$\|\boldsymbol{x}_i - \boldsymbol{x}_j\|_{T_p} = \left\| \frac{i\boldsymbol{z} \bmod n}{n} - \frac{j\boldsymbol{z} \bmod n}{n} \right\|_{T_p} = \left\| \frac{(i-j)\boldsymbol{z} \bmod n}{n} \right\|_{T_p} = \left\| \frac{k\boldsymbol{z} \bmod n}{n} \right\|_{T_p} = \|\boldsymbol{x}_k\|_{T_p},$$
(24)

where $\|\mathbf{x}\|_{T_p}$ denotes the $l_p$-norm-based toroidal distance between $\mathbf{x}$ and $\mathbf{0}$, and $k \equiv i - j \bmod n$.

For non-identical pair $\boldsymbol{x}_i, \boldsymbol{x}_j \in X = \{\boldsymbol{x}_0, \cdots, \boldsymbol{x}_{n-1}\}$, we know $i \neq j$. Thus, $i - j \equiv k \in \{1, \cdots, n-1\}$. Moreover, for each $k$, there are $n$ pairs of $i, j \in \{0, \cdots, n-1\}$ obtaining $i - j \equiv k \bmod n$. Therefore, the non-identical pairwise toroidal distance is determined by $\|\boldsymbol{x}_k\|_{T_p}$ for $k \in \{1, \cdots, n-1\}$. Moreover, each $\|\boldsymbol{x}_k\|_{T_p}$ corresponds to $n$ pairwise distances.

From the definition of the $l_p$-norm-based toroidal distance, we know that

$$\|\boldsymbol{x}_k\|_{T_p} = \left\| \min\left( \frac{k\boldsymbol{z} \bmod n}{n}, \frac{n - k\boldsymbol{z} \bmod n}{n} \right) \right\|_p$$

$$= \left\| \min\left( \frac{k\boldsymbol{z} \bmod n}{n}, \frac{(-k\boldsymbol{z}) \bmod n}{n} \right) \right\|_p,$$
(25)

where $\min(\cdot, \cdot)$ denotes the element-wise min operation between two inputs.

Since $n$ is a prime number, from the number theory, we know that for a primitive root $g$, the residue of $\{g^0, g^1, \cdots, g^{n-2}\}$ modulo $n$ forms a cyclic group under multiplication, and $g^{n-1} \equiv 1 \bmod n$. Moreover, there is a one-to-one correspondence between the residue of $\{g^0, g^1, \cdots, g^{n-2}\}$ modulo $n$ and the set $\{1, 2, \cdots, n-1\}$. Then, we know that $\exists k', g^{k'} \equiv k \bmod n$. It follows that

$$\|\boldsymbol{x}_k\|_{T_p} = \left\| \min\left( \frac{g^{k'}\boldsymbol{z} \bmod n}{n}, \frac{(-g^{k'}\boldsymbol{z}) \bmod n}{n} \right) \right\|_p.$$
(26)

Since $(g^{\frac{n-1}{2}})^2 = g^{n-1} \equiv 1 \bmod n$ and $g$ is a primitive root, we know that $g^{\frac{n-1}{2}} \equiv -1 \bmod n$. Denote $\{\boldsymbol{z}, -\boldsymbol{z}\} := \{z_1, z_2, \cdots, z_d, -z_1, z_2, \cdots, -z_d\}$. Since $\boldsymbol{z} = [g^0, g^{\frac{n-1}{2d}}, g^{\frac{2(n-1)}{2d}}, \cdots, g^{\frac{(d-1)(n-1)}{2d}}] \bmod n$, we know that

$$\{\boldsymbol{z}, -\boldsymbol{z}\} \equiv \{\boldsymbol{z}, g^{\frac{n-1}{2}}\boldsymbol{z}\} \bmod n \tag{27}$$

$$\equiv \{g^0, g^{\frac{n-1}{2d}}, g^{\frac{2(n-1)}{2d}}, \cdots, g^{\frac{(d-1)(n-1)}{2d}}, g^{\frac{n-1}{2}+0}, g^{\frac{n-1}{2}+\frac{n-1}{2d}}, \cdots, g^{\frac{n-1}{2}+\frac{(d-1)(n-1)}{2d}}\} \bmod n \tag{28}$$

$$\equiv \{g^0, g^{\frac{n-1}{2d}}, g^{\frac{2(n-1)}{2d}}, \cdots, g^{\frac{(d-1)(n-1)}{2d}}, g^{\frac{d(n-1)}{2d}}, g^{\frac{(d+1)(n-1)}{2d}}, \cdots, g^{\frac{(2d-1)(n-1)}{2d}}\} \bmod n. \tag{29}$$

It follows that $H := \{z_1, z_2, \cdots, z_d, -z_1, z_2, \cdots, -z_d\} \bmod n$ forms a subgroup of the group $\{g^0, g^1, \cdots, g^{n-2}\} \bmod n$. From Lagrange's theorem in group theory [10], we know that the cosets

of the subgroup $H$ partition the entire group $\{g^0, g^1, \cdots, g^{n-2}\}$ into equal-size, non-overlapping sets, i.e., cosets $g^0 H, g^1 H, \cdots, g^{\frac{n-1-2d}{2d}} H$, and the number of cosets of $H$ is $\frac{n-1}{2d}$.

Together with Eq.(26), we know that distance $\|\boldsymbol{x}_k\|_{T_p}$ for $k' \in \{0, \cdots, n-2\}$ has $\frac{n-1}{2d}$ different values simultaneously hold for all $p \in (0, \infty)$, i.e, $\left\|\min\left(\frac{g^h \boldsymbol{z} \bmod n}{n}, \frac{(-g^h \boldsymbol{z}) \bmod n}{n}\right)\right\|_p$ for $h \in \{0, \cdots, \frac{n-1}{2d}-1\}$. And for each distance value, there are the same number of terms $\|\boldsymbol{x}_k\|_{T_p}$ that obtain this value. Since each $\|\boldsymbol{x}_k\|_{T_p}$ corresponds to $n$ pairwise distance $\|\boldsymbol{x}_i - \boldsymbol{x}_j\|_{T_p}$, where $k \equiv i - j \bmod n$, there are $\frac{n-1}{2d}$ distinct pairwise toroidal distance. Moreover, for each distance value, there are the same number of pairs that obtain this value.

$\square$

## B  Proof of Theorem 2

**Theorem.** *Suppose $n$ is a prime number and $n \geq 2d+1$. Let $\boldsymbol{z} = [z_1, z_2, \cdots, z_d]$ with $1 \leq z_k \leq n-1$. Construct non-degenerate rank-1 lattice $X = \{\boldsymbol{x}_0, \cdots, \boldsymbol{x}_{n-1}\}$ with $\boldsymbol{x}_i = \frac{i\boldsymbol{z} \bmod n}{n}, i \in \{0, ..., n-1\}$. Then, the minimum pairwise toroidal distance can be bounded as*

$$\frac{d(d+1)}{2n} \leq \min_{i,j \in \{0,\cdots,n-1\}, i \neq j} \|\mathbf{x}_i - \mathbf{x}_j\|_{T_1} \leq \frac{(n+1)d}{4n} \tag{30}$$

$$\frac{\sqrt{6d(d+1)(2d+1)}}{6n} \leq \min_{i,j \in \{0,\cdots,n-1\}, i \neq j} \|\mathbf{x}_i - \mathbf{x}_j\|_{T_2} \leq \sqrt{\frac{(n+1)d}{12n}}, \tag{31}$$

*where $\|\cdot\|_{T_1}$ and $\|\cdot\|_{T_2}$ denotes the $l_1$-norm-based toroidal distance and the $l_2$-norm-based toroidal distance, respectively.*

*Proof.* From the definition of the rank-1 lattice, we know that

$$\|\boldsymbol{x}_i - \boldsymbol{x}_j\|_{T_p} = \left\|\frac{i\boldsymbol{z} \bmod n}{n} - \frac{j\boldsymbol{z} \bmod n}{n}\right\|_{T_p} = \left\|\frac{(i-j)\boldsymbol{z} \bmod n}{n}\right\|_{T_p} = \left\|\frac{k\boldsymbol{z} \bmod n}{n}\right\|_{T_p} = \|\boldsymbol{x}_k\|_{T_p}, \tag{32}$$

where $\|\mathbf{x}\|_{T_p}$ denotes the $l_p$-norm-based toroidal distance, we know that between $\mathbf{x}$ and $\mathbf{0}$, and $k \equiv i - j \bmod n$.

Thus, the minimum pairwise toroidal distance is equivalent to Eq. (33)

$$\min_{i,j \in \{0,\cdots,n-1\}, i \neq j} \|\mathbf{x}_i - \mathbf{x}_j\|_{T_p} = \min_{k \in \{1,\cdots,n-1\}} \|\mathbf{x}_k\|_{T_p}. \tag{33}$$

Since the minimum value is smaller than the average value, it follows that

$$\min_{i,j \in \{0,\cdots,n-1\}, i \neq j} \|\mathbf{x}_i - \mathbf{x}_j\|_{T_p} = \min_{k \in \{1,\cdots,n-1\}} \|\mathbf{x}_k\|_{T_p} \leq \frac{\sum_{k=1}^{n-1} \|\boldsymbol{x}_k\|_{T_p}}{n-1}. \tag{34}$$

Since $n$ is a prime number, from number theory, we know that for a primitive root $g$, the residue of $\{g^0, g^1, \cdots, g^{n-2}\}$ modulo $n$ forms a cyclic group under multiplication, and $g^{n-1} \equiv 1 \bmod n$. Moreover, there is a one-to-one correspondence between the residue of $\{g^0, g^1, \cdots, g^{n-2}\}$ modulo $n$ and the set $\{1, 2, \cdots, n-1\}$. Then, for each $t^{th}$ component of $\boldsymbol{z} = [z_1, z_2, \cdots, z_d]$, we know that $\exists m_t$ such that $g^{m_t} \equiv z_t \bmod n$. Therefore, the set $\{kz_t \bmod n | \forall k \in \{1, \cdots, n-1\}\}$ is a permutation of the set $\{1, \cdots, n-1\}$.

From the definition of the $l_p$-norm-based toroidal distance, we know that each $t^{th}$ component of $\|\mathbf{x}_k\|_{T_p}$ is determined by $\min(kz_t \bmod n, n - kz_t \bmod n)$. Because the set $\{kz_t \bmod n | \forall k \in \{1, \cdots, n-1\}\}$ is a permutation of set $\{1, \cdots, n-1\}$, we know that the set $\{\min(kz_t \bmod n, n - kz_t \bmod n) | \forall k \in \{1, \cdots, n-1\}\}$ consists of two copy of permutation of the set $\{1, \cdots, \frac{n-1}{2}\}$. It follows that

$$\sum_{k=1}^{n-1} \|\boldsymbol{x}_k\|_{T_1} = \frac{\sum_{t=1}^d \sum_{k=1}^{n-1} \min(kz_t \bmod n, n - kz_t \bmod n)}{n} = \frac{2d \sum_{k=1}^{\frac{n-1}{2}} k}{n} = \frac{d(n+1)(n-1)}{4n}. \tag{35}$$

Similarly, for $l_2$-norm-based toroidal distance, we have that

$$\sum_{k=1}^{n-1} \|\boldsymbol{x}_k\|_{T_2}^2 = \frac{\sum_{t=1}^{d}\sum_{k=1}^{n-1} \min(kz_t \bmod n, n - kz_t \bmod n)^2}{n^2} = \frac{2d\sum_{k=1}^{\frac{n-1}{2}} k^2}{n^2} = \frac{d(n-1)(n+1)}{12n}.$$
(36)

By Cauchy–Schwarz inequality, we know that

$$\sum_{k=1}^{n-1} \|\boldsymbol{x}_k\|_{T_2} \leq \sqrt{(n-1)\sum_{k=1}^{n-1} \|\boldsymbol{x}_k\|_{T_2}^2} = (n-1)\sqrt{\frac{d(n+1)}{12n}}.$$
(37)

Together with Eq.(34), it follows that

$$\min_{i,j\in\{0,\cdots,n-1\},i\neq j} \|\mathbf{x}_i - \mathbf{x}_j\|_{T_1} = \min_{k\in\{1,\cdots,n-1\}} \|\mathbf{x}_k\|_{T_1} \leq \frac{(n+1)d}{4n}$$
(38)

$$\min_{i,j\in\{0,\cdots,n-1\},i\neq j} \|\mathbf{x}_i - \mathbf{x}_j\|_{T_2} = \min_{k\in\{1,\cdots,n-1\}} \|\mathbf{x}_k\|_{T_2} \leq \sqrt{\frac{(n+1)d}{12n}}.$$
(39)

Now, we are going to prove the lower bound. For a non-degenerate rank-1 lattice, the components of generating vector $\boldsymbol{z} = [z_1, \cdots, z_d]$ should be all different. Then, we know the components of $\boldsymbol{x}_k, \forall k \in \{1, \cdots, n-1\}$ should be all different. Thus, the min norm point is achieved at $\boldsymbol{x}^* = [1/n, 2/n, \cdots, d/n]$. Since $n \geq 2d+1$, it follows that

$$\min_{i,j\in\{0,\cdots,n-1\},i\neq j} \|\mathbf{x}_i - \mathbf{x}_j\|_{T_1} = \min_{k\in\{1,\cdots,n-1\}} \|\mathbf{x}_k\|_{T_1} \geq \|\mathbf{x}^*\|_{T_1} = \frac{(d+1)d}{2n}$$
(40)

$$\min_{i,j\in\{0,\cdots,n-1\},i\neq j} \|\mathbf{x}_i - \mathbf{x}_j\|_{T_2} = \min_{k\in\{1,\cdots,n-1\}} \|\mathbf{x}_k\|_{T_2} \geq \|\mathbf{x}^*\|_{T_2} = \frac{\sqrt{6d(d+1)(2d+1)}}{6n}.$$
(41)

□

## C   Proof of Corollary 1

**Corollary 1.** *Suppose $n = 2d+1$ is a prime number. Let $g$ be a primitive root of $n$. Let $\boldsymbol{z} = [g^0, g^{\frac{n-1}{2d}}, g^{\frac{2(n-1)}{2d}}, \cdots, g^{\frac{(d-1)(n-1)}{2d}}] \bmod n$. Construct rank-1 lattice $X = \{\boldsymbol{x}_0, \cdots, \boldsymbol{x}_{n-1}\}$ with $\boldsymbol{x}_i = \frac{i\boldsymbol{z}\bmod n}{n}, i \in \{0, ..., n-1\}$. Then, the pairwise toroidal distance of the lattice $X$ attains the upper bound.*

$$\|\mathbf{x}_i - \mathbf{x}_j\|_{T_1} = \frac{(n+1)d}{4n}, \forall i,j \in \{0, \cdots, n-1\}, i \neq j,$$
(42)

$$\|\mathbf{x}_i - \mathbf{x}_j\|_{T_2} = \sqrt{\frac{(n+1)d}{12n}}, \forall i,j \in \{0, \cdots, n-1\}, i \neq j.$$
(43)

*Proof.* From the definition of the rank-1 lattice, we know that

$$\|\boldsymbol{x}_i - \boldsymbol{x}_j\|_{T_p} = \left\|\frac{i\boldsymbol{z}\bmod n}{n} - \frac{j\boldsymbol{z}\bmod n}{n}\right\|_{T_p} = \left\|\frac{(i-j)\boldsymbol{z}\bmod n}{n}\right\|_{T_p} = \left\|\frac{k\boldsymbol{z}\bmod n}{n}\right\|_{T_p} = \|\boldsymbol{x}_k\|_{T_p},$$
(44)

where $\|\mathbf{x}\|_{T_p}$ denote the $l_p$-norm-based toroidal distance, we know that between $\mathbf{x}$ and $\mathbf{0}$, and $k \equiv i - j \bmod n$.

From Theorem 1, we know that $\|\boldsymbol{x}_i - \boldsymbol{x}_j\|_{T_p} \forall i,j \in \{0, \cdots, n-1\}, i \neq j$ has $\frac{n-1}{2d}$ different values. Since $n = 2d+1$, we know the pairwise toroidal distance has the same value. Therefore, we know that

$$\|\boldsymbol{x}_i - \boldsymbol{x}_j\|_{T_p} = \|\boldsymbol{x}_k\|_{T_p} = \frac{\sum_{k=1}^{n-1} \|\boldsymbol{x}_k\|_{T_p}}{n-1}, \forall i,j \in \{0, \cdots, n-1\}, i \neq j.$$
(45)

From the proof of Theorem 2, we know that

$$\sum_{k=1}^{n-1} \|\boldsymbol{x}_k\|_{T_1} = \frac{\sum_{t=1}^{d} \sum_{k=1}^{n-1} \min(kz_t \bmod n, n - kz_t \bmod n)}{n} = \frac{2d \sum_{k=1}^{\frac{n-1}{2}} k}{n} = \frac{d(n+1)(n-1)}{4n}. \tag{46}$$

and

$$\sum_{k=1}^{n-1} \|\boldsymbol{x}_k\|_{T_2}^2 = \frac{\sum_{t=1}^{d} \sum_{k=1}^{n-1} \min(kz_t \bmod n, n - kz_t \bmod n)^2}{n^2} = \frac{2d \sum_{k=1}^{\frac{n-1}{2}} k^2}{n^2} = \frac{d(n-1)(n+1)}{12n}. \tag{47}$$

Together Eq.(46) with Eq.(45), we know that

$$\|\mathbf{x}_i - \mathbf{x}_j\|_{T_1} = \frac{(n+1)d}{4n}, \forall i,j \in \{0, \cdots, n-1\}, i \neq j. \tag{48}$$

Since $\|\boldsymbol{x}_1\|_{T_p} = \|\boldsymbol{x}_2\|_{T_p} = \cdots = \|\boldsymbol{x}_{n-1}\|_{T_p}$, it follows that

$$\sum_{k=1}^{n-1} \|\boldsymbol{x}_k\|_{T_2} = \sqrt{(n-1) \sum_{k=1}^{n-1} \|\boldsymbol{x}_k\|_{T_2}^2}. \tag{49}$$

Together with Eq.(47), we know that

$$\sum_{k=1}^{n-1} \|\boldsymbol{x}_k\|_{T_2} = \sqrt{(n-1) \sum_{k=1}^{n-1} \|\boldsymbol{x}_k\|_{T_2}^2} = (n-1)\sqrt{\frac{d(n+1)}{12n}}. \tag{50}$$

Plug Eq.(50) into Eq.(45), if follows that

$$\|\mathbf{x}_i - \mathbf{x}_j\|_{T_2} = \sqrt{\frac{(n+1)d}{12n}}, \forall i,j \in \{0, \cdots, n-1\}, i \neq j. \tag{51}$$

From Theorem 2, we know that the $l_1$-norm-based and $l_2$-norm-based pairwise toroidal distance of the lattice $X$ attains the upper bound.

$\square$

## D   Subgroup-based QMC on Sphere $\mathbb{S}^{d-1}$

In this section, we propose a closed-form subgroup-based QMC method on the sphere $\mathbb{S}^{d-1}$ instead of unit cube $[0,1]^d$. QMC uniformly on sphere can be used to construct samples for isotropic distribution, which is helpful for variance reduction of the gradient estimators in Evolutionary strategy for reinforcement learning [30].

Lyu [23] constructs structured sampling matrix on $\mathbb{S}^{d-1}$ by minimizing the discrete Riesz energy. In contrast, we construct samples by a closed-form construction without the time-consuming optimization procedure. Our construction can achieve a small mutual coherence.

Without loss of generality, we assume that $d = 2m$, $N = 2n$, and $n$ is a prime such that $m|(n-1)$. Let $F \in \mathbb{C}^{n \times n}$ be a $n \times n$ discrete Fourier matrix. $F_{k,j} = e^{\frac{2\pi ikj}{n}}$ is the $(k,j)^{th}$ entry of $F$, where $i = \sqrt{-1}$. Let $\Lambda = \{k_1, k_2, ..., k_m\} \subset \{1, ..., n-1\}$ be a subset of indexes.

The structured sampling matrix $\mathbf{V}$ in [23] can be defined as equation (52).

$$\mathbf{V} = \frac{1}{\sqrt{m}} \begin{bmatrix} \mathrm{Re}F_\Lambda & -\mathrm{Im}F_\Lambda \\ \mathrm{Im}F_\Lambda & \mathrm{Re}F_\Lambda \end{bmatrix} \in \mathbb{R}^{d \times N} \tag{52}$$

where Re and Im denote the real and image part of a complex number, and $F_\Lambda$ in equation (53) is the matrix constructed by $m$ rows of $F$

$$F_\Lambda = \begin{bmatrix} e^{\frac{2\pi i k_1 1}{n}} & \cdots & e^{\frac{2\pi i k_1 n}{n}} \\ \vdots & \ddots & \vdots \\ e^{\frac{2\pi i k_m 1}{n}} & \cdots & e^{\frac{2\pi i k_m n}{n}} \end{bmatrix} \in \mathbb{C}^{m \times n}. \tag{53}$$

With the $\mathbf{V}$ given in equation (52), we know that $\|\boldsymbol{v}_i\|_2 = 1$ for $i \in \{1, ..., n\}$. Thus, each column of matrix $\mathbf{V}$ is a point on $\mathbb{S}^{d-1}$.

Let $g$ denote a primitive root modulo $n$. We construct the index $\Lambda = \{k_1, k_2, ..., k_m\}$ as

$$\Lambda = \{g^0, g^{\frac{n-1}{m}}, g^{\frac{2(n-1)}{m}}, \cdots, g^{\frac{(m-1)(n-1)}{m}}\} \bmod n. \tag{54}$$

The set $\{g^0, g^{\frac{n-1}{m}}, g^{\frac{2(n-1)}{m}}, \cdots, g^{\frac{(m-1)(n-1)}{m}}\}$ mod $n$ forms a subgroup of the the group $\{g^0, g^1, \cdots, g^{n-2}\}$ mod $n$. Based on this, we derive upper bounds of the mutual coherence of the points set $\mathbf{V}$. The results are summarized in Theorem 3 and Theorem 4.

**Theorem 3.** *Suppose $d = 2m, N = 2n$, and $n$ is a prime such that $m|(n-1)$. Construct matrix $\mathbf{V}$ as in Eq.(52) with index set $\Lambda$ as Eq.(54). Let mutual coherence $\mu(\mathbf{V}) := \max_{i \neq j} |\boldsymbol{v}_i^\top \boldsymbol{v}_j|$. Then $\mu(\mathbf{V}) \leq \frac{\sqrt{n}}{m}$.*

**Theorem 4.** *Suppose $d = 2m, N = 2n$, and $n$ is a prime such that $m|(n-1)$, and $m \leq n^{\frac{2}{3}}$. Construct matrix $\mathbf{V}$ as in Eq.(52) with index set $\Lambda$ as Eq.(54). Let mutual coherence $\mu(\mathbf{V}) := \max_{i \neq j} |\boldsymbol{v}_i^\top \boldsymbol{v}_j|$. Then $\mu(\mathbf{V}) \leq Cm^{-1/2}n^{1/6}\log^{1/6} m$, where $C$ denotes a positive constant independent of $m$ and $n$.*

Theorem 3 and Theorem 4 show that our construction can achieve a bounded mutual coherence. A smaller mutual coherence means that the points are more evenly spread on sphere $\mathbb{S}^{d-1}$.

**Remark:** Our construction does not require a restrictive constraint of the dimension of data. The only assumption of data dimension $d$ is that $d$ is a even number, i.e.,$2|d$, which is commonly satisfied in practice. Moreover, the product $\mathbf{V}^\top \boldsymbol{x}$ can be accelerated by fast Fourier transform as in [23].

### D.1 Evaluation of the mutual coherence

We evaluate our subgroup-based spherical QMC by comparing with the construction in [23] and i.i.d Gaussian sampling.

We set the dimension $d$ as in $\{50, 100, 200, 500, 1000\}$. For each dimension $d$, we set the number of points $N = 2n$, with $n$ as the first ten prime numbers such that $\frac{d}{2}$ divides $n-1$, i.e., $\frac{d}{2}|(n-1)$. Both subgroup-based QMC and Lyu's method are deterministic. For Gaussian sampling method, we report the mean $\pm$ standard deviation of mutual coherence over 50 independent runs. The mutual coherence for each dimension are reported in Table 3. The smaller the mutual coherence, the better.

We can observe that our subgroup-based spherical QMC achieves a competitive mutual coherence compared with Lyu's method in [23]. Note that our method does not require a time consuming optimization procedure, thus it is appealing for applications that demands a fast construction. Moreover, both our subgroup-based QMC and Lyu's method obtain a significant smaller coherence than i.i.d Gaussian sampling.

Table 3: Mutual coherence of points set constructed by different methods. Smaller is better.

| d=50 | | 202 | 302 | 502 | 802 | 1202 | 1402 | 1502 | 2102 | 2302 | 2402 |
|---|---|---|---|---|---|---|---|---|---|---|---|
| | SubGroup | **0.1490** | **0.2289** | **0.1923** | 0.2930 | **0.2608** | 0.3402 | 0.3358 | 0.3211 | 0.4534 | **0.3353** |
| | Lyu [23] | 0.2313 | 0.2377 | 0.2901 | **0.2902** | 0.3005 | **0.3154** | **0.3155** | **0.3209** | **0.3595** | 0.3718 |
| | Gaussian | 0.5400± 0.0254 | 0.5738± 0.0291 | 0.5904± 0.0257 | 0.6158± 0.0249 | 0.6270± 0.0209 | 0.6254± 0.0184 | 0.6328± 0.0219 | 0.6447± 0.0184 | 0.6520± 0.0204 | 0.6517± 0.0216 |

| d=100 | | 202 | 302 | 502 | 802 | 1202 | 1402 | 1502 | 2102 | 2302 | 2402 |
|---|---|---|---|---|---|---|---|---|---|---|---|
| | SubGroup | **0.1105** | **0.1529** | 0.1923 | **0.1764** | 0.2397 | 0.2749 | 0.2513 | 0.2679 | 0.4534 | 0.3353 |
| | Lyu [23] | 0.1234 | 0.1581 | **0.1586** | 0.1870 | **0.2041** | **0.2191** | **0.1976** | **0.2047** | **0.2244** | **0.2218** |
| | Gaussian | 0.4033± 0.0272 | 0.4210± 0.0274 | 0.4422± 0.0225 | 0.4577± 0.0230 | 0.4616± 0.0170 | 0.4734± 0.0174 | 0.4716± 0.0234 | 0.4878± 0.0167 | 0.4866± 0.0172 | 0.4947± 0.0192 |

| d=200 | | 202 | 802 | 1202 | 1402 | 2402 | 2602 | 3202 | 3602 | 3802 | 5602 |
|---|---|---|---|---|---|---|---|---|---|---|---|
| | SubGroup | **0.0100** | 0.1251 | 0.1835 | 0.1966 | 0.2365 | 0.1553 | 0.1910 | 0.1914 | 0.2529 | 0.2457 |
| | Lyu [23] | **0.0100** | **0.1108** | **0.1223** | **0.1262** | **0.1417** | **0.1444** | **0.1505** | **0.1648** | **0.1624** | **0.1679** |
| | Gaussian | 0.2887± 0.0163 | 0.3295± 0.0155 | 0.3362± 0.0148 | 0.3447± 0.0182 | 0.3564± 0.0140 | 0.3578± 0.0142 | 0.3645± 0.0143 | 0.3648± 0.0142 | 0.3689± 0.0140 | 0.3768± 0.0151 |

| d=500 | | 502 | 1502 | 4502 | 6002 | 6502 | 8002 | 9502 | 11002 | 14002 | 17002 |
|---|---|---|---|---|---|---|---|---|---|---|---|
| | SubGroup | **0.0040** | 0.0723 | 0.1051 | 0.1209 | 0.1107 | 0.1168 | 0.1199 | 0.1425 | 0.1587 | 0.1273 |
| | Lyu [23] | **0.0040** | **0.0650** | **0.0946** | **0.0934** | **0.0930** | **0.1004** | **0.0980** | **0.1022** | **0.1077** | **0.1110** |
| | Gaussian | 0.2040± 0.0111 | 0.2218± 0.0099 | 0.2388± 0.0092 | 0.2425± 0.0081 | 0.2448± 0.0113 | 0.2498± 0.0110 | 0.2528± 0.0100 | 0.2527± 0.0084 | 0.2579± 0.0113 | 0.2607± 0.0092 |

| d=1000 | | 6002 | 8002 | 11002 | 14002 | 17002 | 18002 | 21002 | 26002 | 32002 | 38002 |
|---|---|---|---|---|---|---|---|---|---|---|---|
| | SubGroup | 0.0754 | 0.0778 | 0.0819 | 0.0921 | 0.0935 | 0.0764 | 0.1065 | 0.0931 | 0.0908 | 0.1125 |
| | Lyu [23] | 0.0594 | 0.0637 | 0.0662 | 0.0680 | 0.0684 | 0.0744 | 0.0774 | 0.0815 | 0.0781 | 0.0814 |
| | Gaussian | 0.1736± 0.0067 | 0.1764± 0.0059 | 0.1797± 0.0060 | 0.1828± 0.0062 | 0.1846± 0.0052 | 0.1840± 0.0057 | 0.1869± 0.0052 | 0.1888± 0.0055 | 0.1909± 0.0067 | 0.1920± 0.0056 |

# E  Proof of Theorem 3

*Proof.* Let $c_i \in \mathbb{C}^m$ be the $i^{th}$ column of matrix $F_\Lambda \in \mathbb{C}^{m \times n}$ in Eq.(53). Let $v_i \in \mathbb{R}^{2m}$ be the $i^{th}$ column of matrix $V \in \mathbb{R}^{2m \times 2n}$ in Eq.(52). For $1 \leq i, j \leq n, i \neq j$, we know that

$$v_i^\top v_{i+n} = 0, \tag{55}$$

$$v_{i+n}^\top v_{j+n} = v_i^\top v_j = \mathrm{Re}(c_i^* c_j), \tag{56}$$

$$v_{i+n}^\top v_j = -v_i^\top v_{j+n} = \mathrm{Im}(c_i^* c_j), \tag{57}$$

where $*$ denote the complex conjugate, $\mathrm{Re}(\cdot)$ and $\mathrm{Im}(\cdot)$ denote the real and image part of the input complex number.

It follows that

$$\mu(V) \leq= \max_{1 \leq k, r \leq 2n, k \neq r} |v_k^\top v_r| \leq \max_{1 \leq i, j \leq n, i \neq j} |c_i^* v_j| = \mu(F_\Lambda) \tag{58}$$

From the definition of $F_\Lambda$ in Eq.(53), we know that

$$\mu(F_\Lambda) = \max_{1 \leq i, j \leq n, i \neq j} |c_i^* v_j| = \max_{1 \leq i, j \leq n, i \neq j} \frac{1}{m} \left| \sum_{z \in \Lambda} e^{\frac{2\pi i z (j-i)}{n}} \right| \tag{59}$$

$$= \max_{1 \leq k \leq n-1} \frac{1}{m} \left| \sum_{z \in \Lambda} e^{\frac{2\pi i z k}{n}} \right| \tag{60}$$

Because $\Lambda$ is a subgroup of the multiplicative group $\{g^0, g^1, \cdots, g^{n-2}\}$ mod $n$, from [4], we know that

$$\max_{1 \leq k \leq n-1} \left| \sum_{z \in \Lambda} e^{\frac{2\pi i z k}{n}} \right| \leq \sqrt{n} \tag{61}$$

Finally, we know that

$$\mu(V) \leq \mu(F_\Lambda) \leq \frac{\sqrt{n}}{m}. \tag{62}$$

□

# F Proof of Theorem 4

*Proof.* Because $\Lambda$ is a subgroup of the multiplicative group $\{g^0, g^1, \cdots, g^{n-2}\}$ mod $n$, and $m \leq n^{2/3}$, from Theorem 1 in [31], we know that

$$\max_{1 \leq k \leq n-1} \left| \sum_{z \in \Lambda} e^{\frac{2\pi i z k}{n}} \right| \leq C m^{1/2} n^{1/6} \log^{1/6} m \tag{63}$$

From the proof of Theorem 3, we have that

$$\mu(V) \leq \mu(F_\Lambda) = \max_{1 \leq k \leq n-1} \frac{1}{m} \left| \sum_{z \in \Lambda} e^{\frac{2\pi i z k}{n}} \right| \leq C m^{-1/2} n^{1/6} \log^{1/6} m \tag{64}$$

$\square$

# G QMC for Generative models

Our subgroup rank-1 lattice can be used for generative models. Buchholz et al. [5] suggest using QMC for variational inference to maximize the evidence lower bound (ELBO). We present a new method by directly learning the inverse of the cumulative distribution function (CDF).

In variational autoencoder, the objective is the evidence lower bound (ELBO) [15] defined as

$$\mathcal{L}(x, \phi, \theta) = \mathbb{E}_{q_\phi(z|x)} \left[ \log p_\theta(x|z) \right] - \text{KL} \left[ q_\phi(z|x) || p_\theta(z) \right]. \tag{65}$$

The ELBO consists of two terms, i.e., the reconstruction term $\mathbb{E}_{q_\phi(z|x)} \left[ \log p_\theta(x|z) \right]$ and the regularization term $\text{KL} \left[ q_\phi(z|x) || p_\theta(z) \right]$. The reconstruction term is learning to fit, while the regularization term controls the distance between distribution $q_\phi(z|x)$ to the prior distribution $p_\theta(z)$.

The reconstruction term $\mathbb{E}_{q_\phi(z|x)} \left[ \log p_\theta(x|z) \right]$ can be reformulated as

$$\mathbb{E}_{q_\phi(z|x)} \left[ \log p_\theta(x|z) \right] = \int_{\mathcal{Z}} q_\phi(z|x) \log p_\theta(x|z) \mathrm{d}\boldsymbol{z} \tag{66}$$

$$= \int_{[0,1]^d} \log p_\theta \left( x | \Phi^{-1}(\boldsymbol{\epsilon}) \right) \mathrm{d}\boldsymbol{\epsilon}. \tag{67}$$

where $\Phi^{-1}(\cdot)$ denotes the inverse cumulative distribution function with respect to the density $q_\phi(z|x)$.

Eq.(67) provides an alternative training scheme, we directly learn the inverse of CDF $F(\boldsymbol{\epsilon}; x) = \Phi^{-1}(\boldsymbol{\epsilon})$ given $x$ instead of the density $q_\phi(z|x)$. We parameterize $F(\boldsymbol{\epsilon}, x)$ as a neural network with input $\boldsymbol{\epsilon}$ and data $x$. The inverse of CDF function $F(\boldsymbol{\epsilon}, x)$ can be seen as an encoder of $x$ for inference. It is worth noting that learning the inverse of CDF can bring more flexibility without the assumption of the distribution, e.g., Gaussian.

To ensure the distribution $q$ close to the prior distribution $p(z)$, we can use other regularization terms instead of the KL-divergence for any implicit distribution $q$, e.g., the maximum mean discrepancy. Besides this, we can also use a discriminator-based adversarial loss similar to adversarial autoencoders [24]

$$\widetilde{L}(x, F, D) = \mathbb{E}_{p_\theta(\boldsymbol{z})} \left[ \log(D(\boldsymbol{z})) \right] + \mathbb{E}_{p(\boldsymbol{\epsilon})} \left[ \log(1 - D(F(\boldsymbol{\epsilon}, x))) \right], \tag{68}$$

where $p(\boldsymbol{\epsilon})$ denotes a uniform distribution on unit cube $[0, 1]^d$, $D$ is the discriminator, $F$ denotes the inverse of CDF mapping.

When the domain $\mathcal{Z}$ coincides with a target domain $\mathcal{Y}$, we can use an empirical data distribution $Y$ as the prior. This leads to a training scheme similar to cycle GAN [36]. In contrast to cycle GAN, the encoder $F$ depends on both data $x$ in source domain and $\boldsymbol{\epsilon}$ in unit cube. The expectation term $\mathbb{E}_{p(\boldsymbol{\epsilon})}[\cdot]$ can be approximated by QMC methods.

# H  Generative Inference for CycleGAN

We evaluate our subgroup rank-1 lattice on training generative model. As shown in section G, we can learn the inverse CDF functions $F(\epsilon, x)$ as a generator from domain $\mathcal{X}$ to domain $\mathcal{Y}$ in cycle GAN. We set $F(\epsilon, x) = G_1(x) + G_2(\epsilon)$, where $G_1$ and $G_2$ denotes the neural networks. Network $G_1$ maps input image $x$ to a target mean, while network $G_2$ maps $\epsilon \in [0, 1]^d$ as the residue. Similarly, $\widetilde{F}(\widetilde{\epsilon}, y) = \widetilde{G}_1(y) + \widetilde{G}_2(\widetilde{\epsilon})$ denotes an generator from domain $\mathcal{Y}$ to domain $\mathcal{X}$.

We implement the model based on the open-sourced Pytorch code of [36]. All $G_1$, $G_2$, $\widetilde{G}_1$ and $\widetilde{G}_2$ employ the default ResNet architecture with 9 blocks in [36]. The input size of both $\epsilon$ and $\widetilde{\epsilon}$ are $d = 256 \times 256$. We keep all the hyperparameters same for all the methods as the default value in [36].

We compare our subgroup rank-1 lattice with Monte Carlo sampling for training the generative model. For subgroup rank-1 lattice, we set the number of points $n = 12d + 1 = 786433$. We do not store all the points, instead we sample $i \in \{0, \cdots, n-1\}$ uniformly and construct $\epsilon$ and $\widetilde{\epsilon}$ based on Eq.(3) during the training process. For Monte Carlo sampling, $\epsilon$ and $\widetilde{\epsilon}$ are sampled from $Uniform[0, 1]^d$.

We train generative models on the Vangogh2photo data set and maps data set employed in [36]. We present experimental results of the generated images from models trained with subgroup-based rank-1 lattice sampling, Monte-Carlo sampling, and standard version of CycleGAN. The experimental results on Vangogh2photo dataset and maps dataset are shown in Figure 5 and Figure 6, respectively. From Figure 5, we can observe that the images generated by the model trained with Monte-Carlo sampling have some blurred patches. This phenomenon may be because the additional flexibility of randomness makes the training more difficult to converge to a good model. In contrast, the model trained with subgroup-based rank-1 lattice sampling generates more clearer images. It may be because the rank-1 lattice sampling has finite possible choices, i.e., $n = 786433$ possible points in the experiments, which is much smaller than the case of Monte-Carlo uniform sampling. The rank-1 lattice sampling is more deterministic than Monte Carlo sampling, which alleviates the training difficulty to fit a good model. Since in our subgroup-based rank-1 lattice it is very simple to construct new samples, it can serve as a good alternative to Monte Carlo sampling for generative model training.

Figure 5: Illustration of the generated images from models trained with subgroup rank-1 lattice sampling, Monte-Carlo sampling, and Standard version of CycleGAN.

Figure 6: Illustration of the generated images from models trained with subgroup rank-1 lattice sampling, Monte-Carlo sampling, and Standard version of CycleGAN.

(a) $\frac{\left\|\widetilde{K}-K\right\|_F}{\|K\|_F}$ for Gaussian Kernel (b) $\frac{\left\|\widetilde{K}-K\right\|_F}{\|K\|_F}$ for First-order Arc Kernel (c) $\frac{\left\|\widetilde{K}-K\right\|_F}{\|K\|_F}$ for Zero-order Arc Kernel

(d) $\frac{\left\|\widetilde{K}-K\right\|_\infty}{\|K\|_\infty}$ for Gaussian Kernel (e) $\frac{\left\|\widetilde{K}-K\right\|_\infty}{\|K\|_\infty}$ for First-order Arc Kernel (f) $\frac{\left\|\widetilde{K}-K\right\|_\infty}{\|K\|_\infty}$ for Zero-order Arc Kernel

Figure 7: Relative Mean and Max Reconstruction Error for Gaussian, Zero-order and First-order Arc-cosine Kernel on SIFT1M dataset.