[Reviews · NeurIPS 2020]

Review 1

Summary and Contributions: ***EDIT AFTER REBUTTAL*** I have read the rebuttal and thank the authors for their response and the additional experiments. After careful consideration and discussion with the other reviewers, I have decided to leave my score unchanged. The theoretical component of the paper is strong and the method is a valuable improvement over existing lattice-based QMC methods. However, the experiments have not sufficiently convinced me of the usefulness of the integration method for applications relevant for the NeurIPS community. In their rebuttal, the authors mention a list of potential applications, but it would be a lot more convincing to study such examples explicitly in the paper. In case the paper gets accepted, it would be highly desirable have the grammatical errors in the paper corrected for improved readability. *** The authors propose a closed-form way to construct a rank-1 lattice for Quasi Monte Carlo (QMC), which has previously been done by exhaustive search. Their construction of a generating vector relies on number theoretic insights. The resulting lattice has provable properties in terms of spacing and achieves similar results to the exhaustive search at a much lower computational effort.

Strengths: The work finds a practicable closed-form expression for an approach that previously required exhaustive search. The claims are theoretically grounded and show promising behavior in empirical evaluation. This seems to be a considerable improvement when it comes to rank-1 lattices for QMC.

Weaknesses: Integration experiments could be more diverse, the current setup is an exponential function and as such a well-behaved function. An evaluation of a more diverse set of benchmarks would be desirable.

Correctness: The derivations in the paper appear correct, but I did not check the proofs in the appendix. In the empirical evaluation I wonder why the kernel approximation has been compared to Hua also, while the integral approximation has not. What is the computational requirement of the method by Hua? Could you elaborate more on the computational load for the methods (quite vague in l.210/211)?

Clarity: - The paper is overall well structured and contains all relevant details. However, at times the grammatical structure of sentences hamper understanding. Furthermore, cross-references to relevant sections in the supplementary material are missing, which would greatly enhance the reading experience. E.g., there should be a reference in the main paper to any proof in the appendix, and ideally to any further information that is provided in the appendix. - In 3.1 a motivation of the chosen construction of a generating vector would be helpful to follow the derivation. What makes the specific choice desirable? What is the intuition behind the approach? For me, the derivation came a bit out of nowhere. - The authors should also reassess grammar and spelling for better readability. - The broader impact section is still relatively technical, the authors might to think about effects their algorithm might have further down the road.

Relation to Prior Work: Yes, the authors discuss previous work in the domain of QMC.

Reproducibility: Yes

Additional Feedback: **Comments/Questions:** - eq. (3) and (10) are the same, aren't they? Do you actually need to define the angled brackets at all? - Table 1 and Table 2: please add $n$ to the top row. - Eq. 20 what is $\Phi$? **Typos and grammatical errors:** Please carefully check specifically for articles. Here's a non-exhaustive lists of errors I found in the main paper. - all over the paper: Sobol' should be Sobol without apostrophe afaik - l.17 random feature _maps_ - l.25 either iid or i.i.d. - l.27 i.e., digital nets/sequences and lattice rules - l.30 Korobov introduced - l.32 According to general lattice rules... (or is there just one rule?) - l.36 non-efficient -> inefficient - l.37 non-uniform what? - l.38 are rank-1 lattices? - l.40/41 grammar? - l.43 rank-1 lattice_s_ - l.58 wrt or w.r.t. - l.67 ...does not require the time-consuming exhaustive computer search that previous rank-1 lattice algorithms rely on. - l.92 denotes the operation of taking - l.96 set X in _the_ unit cube - l.107 Define the covering radius... - l.118 ... construction of _a_ rank-1 lattice based on subgroup theory. - l.131 are integer components - l.133 maximize -> maximizing - l.135 set -> setting - l.149 Construct _a_ rank-1... - l.150 distance values - l.151 and each distance value is taken by the same number of pairs in X - l.154 Our Rank-1 Lattice? - l.155 what do you mean by "regular property"? also, _our_ rank-1 lattice, or rank-1 lattices if you consider them more generally - l.157 Construct _a_ rank-1 lattice - l.162 _The_ term - l.207 that _our_ subgroup rank-1 lattice obtains _a_ competitive distance - l.215 We compare with Halton and Sobol sequences. For both we use... - l.225 The test the kernel approximation for _the_ Gaussian kernel, _the_ zeroth-order arc-cosine, etc. - l.227 _a_ Hua rank-1 lattice - l.230 on _the_ DNA and _the_ SIFT1M dataset - l.231 random samples - l.240 has _low_ pairwise distances - l.244 _our_ subgroup... - l.245 CycleGAN training


Review 2

Summary and Contributions: Quasi Monte Carlo (QMC) methods provide an alternative to standard Monte Carlo methods for problems such as integral approximation. This work focuses on the rank-1 lattice QMC formulation due to that formulation's convenient mathematical and computation properties. The author(s) provide an efficient algorithm for selecting a set of points that achieves separation similar to more much computationally-expensive alternatives. The author(s) provide theoretical bounds for their methods and provide an empirical evaluation in terms of minimum pairwise toroidal distance and approximation errors.

Strengths: QMC via lattices is an intuitive and reasonable approach. The author(s) communicate the ideas clearly. The author(s)'s method is 1-2 orders of magnitude faster than exhaustive methods such as Korobov's approach yet it achieves comparable minimum point separation. The execution time improvements increase with dimension and/or the number of points without a similar degradation in point separation. In contrast to previous work, the author(s)'s method does not require verification that generated points are within a valid space. The author(s) prove theoretical bounds on the $\ell_1$ and $\ell_2$ pairwise toroidal distances given specific relationships between the number of samples $n$ and dimension $d$. The author(s) also provide a case where their method converges to the optimal toroidal distance. The author(s)'s proposed method is applicable to any domain where QMC is applicable including integral or kernel approximation.

Weaknesses: The potential weakness of any lattice-based method are clear, and the completeness of the paper would be improved if those were discussed and experimental evaluation provided more of a stress-test of those how well the method compares to baselines in those more deleterious settings. The authors strive to demonstrate the relevance of their work to the NeurIPS community by linking it to more traditional ML tasks (e.g., kernels and GANs). I remain somewhat unconvinced that this NeurIPS is the optimal venue for this type of work. Regardless, inclusion of this paper would add to the breadth and richness of the conference.

Correctness: Experiment methodology in the main paper is reasonable. They compare their results against existing work in three different settings. They also provide source code of their implementation. From a visual check, their implementation was reasonable, well-organized, easy to follow, and without any obvious errors. For their supplemental CycleGAN experiments, I do not believe source code was provided. The CycleGAN experiments that the authors could have compared to more similar previous work. That would have made for a more insightful comparison since it would have clarified the extent to which the improvements were specifically attributable to their method. For Eq. (5), please verify that $\mathcal{X}$ is explicitly defined in the main paper (I only saw it was defined in the supplemental).

Clarity: The paper is very well written. Exposition is clear; the paper is well organized, and the key ideas were readily graspable despite this work not being one of my primary research areas. The organization of writing is a major strength of this work. Given the importance of Theorem 1, I believe the clarity of its sentence should be improved. Other reviewers note multiple grammatical/language issues in the paper. For some of those points, versions of those language issues also appear in the author rebuttal. I am concerned whether these language issues will be sufficiently addressed in a final version.

Relation to Prior Work: The author(s) direct the reader to Dick et al. [7] and Niederreiter [24] for a more complete discussion of QMC and its relation to rank-1 lattice. The paper would benefit from a more complete discussion of specifically how their method contrasts with existing work (in particular Hua & Wang [13] which is only introduced in Section 5). To make the paper more self-contained, it is hoped that this aspect could be improved in the camera-ready version when space is somewhat less constrained.

Reproducibility: Yes

Additional Feedback: Overall I believe this paper is very borderline. Language issues, limited fitness for the venue, and questions about the experiments are shortcomings of the paper. The paper's theoretical contribution is clever. * I appreciate the authors providing an execution time comparison between their method and Korobov in the author response. As the authors stated in their submission, their algorithm is *significantly* faster with the speedup increasing with problem difficulty. I believe that the table be included in future version(s) of the paper for completeness. * Comparison to Korobov in Sec. 5.2: Fig.(a)-(d) in the rebuttal show a clear gap with respect to Hua. However, the y-axis scale makes comparison to MC and Korobov difficult. Perhaps a logarithmic y-axis would better illustrate the differences? The broader impacts section focused primarily on the applications of this work to existing technology domains. In the NeurIPS FAQ, the referenced material recommended that authors also discuss the societal impacts of their work. While I expect there are little societal risks here, even a sentence from the authors indicating they considered such factors and explaining why they did not apply is important. To be clear, this concern did not factor into my scoring and is more general.


Review 3

Summary and Contributions: This paper makes a contribution to lattice-based quasi-Monte Carlo (QMC) by devising an efficient construction procedure for a rank-1 lattice over a bounded set, namely the unit cube. A rank-1 lattice is obtained by repeatedly transforming a single base vector by an invertible transformation and avoids the "quasi-rejection sampling" associated with general lattice rule construction, i.e. the need to repeatedly check if points leave the desired domain, but is still generally difficult because searching for an appropriate generating vector is nontrivial. The authors present a method of circumventing the search by framing the objective as that of finding a lattice with maximum packing radius, and heuristically achieving this objective via a number theoretic argument in Section 3.1, the key part of the paper. In particular, they show that if the generating vector is chosen as specified in Eq (14), the number of distinct pairwise distances in the lattice is minimized, a strategy which they argue typically yields large minimum pairwise distance, i.e. large packing radius, where the metric is defined to be the Lp toroidal distance. More formally, Theorem 1 states that the construction yields a lattice with 1. few unique pairwise distancer values and 2. a "flat histogram" of pairs attaining each value. This is used as a surrogate for lattice regularity. Results comparing the resultant minimum distances with those of other rank-1 lattice construction methods are presented, and the errors of an integration test problem are reported. The overall aim of this class of methods is the determination of a set of points over which one could approximate an integrand more efficiently (i.e. with lower error for a given number of function evaluations) than naive iid Monte Carlo sampling.

Strengths: On the positive side, I believe the technical aspects of this paper are correct, and indeed the procedure proposed for generating the generating vector so as to minimize the number of distinct pairwise distances is rather elegant. I very much enjoyed learning about the relevant number theoretic concepts so as to be able to follow the presented arguments. I think this paper is a positive contribution to the body of research of QMC methods.

Weaknesses: My main concerns are whether this work makes an impactful contribution to the type of problems of interest to the NeurIPS community. The problem of high-dimensional integration is of course of vital importance in many areas of machine learning, appearing centrally for example in Bayesian inference/model selection, graphical models, and the training of latent variable generative models, and many of us would welcome an addition to the toolkit of dealing with such beasts. Unfortunately, this paper makes only minimal effort to motivate the relevance of the proposed QMC construction to these settings. An application to GAN/VAEs does briefly appear in the supplementary, but with quite cursory quantification of performance; showing sharper generated images is not consistent with the rigorous aims and tone of the paper. For a NeurIPS audience, I consider it essential to include a comparison against established sampling algorithms such as Sequential Monte Carlo. A convincing case would be made by showing that this method can achieve performance equivalent to or exceeding that of a more expensive approach on for example, calculating the marginal likelihood of a latent variable model or computing the log partition function of a undirected model in one of the cases in which it can be determined exactly. For what it's worth, I am quite skeptical that the admittedly clever construction proposed in this paper can fundamentally help with the problems which I mentioned above. The ubiquitous issue with statistically-derived high dimensional integrals is that the essential support usually lies on an extremely tiny fraction of the overall domain volume. That is why it is typically essential to include a dynamical aspect to the sampling (e.g. MCMC) to locate the important regions for such problems. A fixed grid that, however nice its properties, disregards the target function seems unlikely to work well due to the fact that it will probably miss these regions. In summary, this paper presents a sound and efficient algorithm for generating a lattice for high-dimensional QMC, its impact on the majority of high dimensional integration problems of interest in machine learning and statistics is unclear and relatively unexplored. It may be better suited to a venue more specialized to QMC work.

Correctness: The claims and methodology appear to my understanding to be correct, but as stated above, do not place the results in the broader context of Monte Carlo methods in widespread use.

Clarity: While the paper is to my understanding technically correct, it contains many grammatical errors and needs to be passed by a proofreader versed with English grammar before being considered publishable.

Relation to Prior Work: Relation to prior work appears to be adequately discussed.

Reproducibility: Yes

Additional Feedback:

[Author Response · NeurIPS 2020]



(a) 50-D      (b) 100-D      (c) 500-D      (d) 1000-D

(e) 10-D      (f) 50-D      (g) 100-D      (h) 500-D

We thank all the reviewers for their constructive comments! We will modify the paper by correcting all grammatical
errors, adding the cross-references and the discussion of Hua's method.

**Q1. (R1 & R2) Comparison of our subgroup rank-1 lattice with Hua and Korobov searching method on integral**
**approximation problem in sec.5.2.** The approximation errors are shown in Fig. (a)-(d). Hua's method obtains a
smaller error than i.i.d Monte Carlo on the 50-D problem, however, it becomes worse than MC on 500-D and 1000-D
problems. Our subgroup rank-1 lattice achieves similar performance to Korobov searching method and obtains a
consistent smaller error on all the tested problems than Hua and MC.

**Q2. (R2) Time Comparison of Korobov searching and our sub-group rank-1 lattice.** The table below shows the
time cost (seconds) for lattice construction. The run time for Korobov searching grows fast to hours. Our method can
run in less than one second, achieving a $10^4\times$ to $10^5\times$ speed-up. The speed-up increases when $n$ and $d$ becomes larger.

| | | n=3001 | 4001 | 7001 | 9001 | 13001 | 16001 | 19001 | 21001 | 24001 | 28001 |
|---|---|---|---|---|---|---|---|---|---|---|---|
| d=500 | SubGroup | 0.0185 | 0.0140 | 0.0289 | 0.043 | 0.0386 | 0.0320 | 0.0431 | 0.0548 | 0.0562 | 0.0593 |
| | Korobov | 34.668 | 98.876 | 152.86 | 310.13 | 624.56 | 933.54 | 1308.9 | 1588.5 | 2058.5 | 2815.9 |
| | | n=4001 | 16001 | 24001 | 28001 | 54001 | 70001 | 76001 | 88001 | 90001 | 96001 |
| d=1000 | SubGroup | 0.0388 | 0.0618 | 0.1041 | 0.1289 | 0.2158 | 0.2923 | 0.3521 | 0.4099 | 0.5352 | 0.5663 |
| | Korobov | 112.18 | 1849.4 | 4115.9 | 5754.6 | 20257 | 34842 | 43457 | 56798 | 56644 | 69323 |

**Q3. (All) More experiments: Approximation of the normalization constant of graphical model.**

For Boltzmann Machines with continuous state in $[0,1]$, the energy function of $\boldsymbol{x}=[\boldsymbol{v},\boldsymbol{h}]\in[0,1]^d$ is defined as
$E(\boldsymbol{x})=-(\boldsymbol{x}^\top\boldsymbol{W}\boldsymbol{x}+\boldsymbol{b}^\top\boldsymbol{x})/d$. The normalization constant is $Z=\int_{[0,1]^d}\exp\left(-E(\boldsymbol{x})\right)d\boldsymbol{x}$.

We evaluate our method on approximation of the normalization constant by comparing with i.i.d Monte Carlo (MC),
slice sampling (SS) and Hamiltonian Monte Carlo (HMC). We generate the elements of $\boldsymbol{W}$ and $\boldsymbol{b}$ by sampling from
standard Gaussian $\mathcal{N}(0,1)$. All the methods in comparison use the same $\boldsymbol{W}$ and $\boldsymbol{b}$. For SS and HMC, we use the
*slicesample* function and *hmcSampler* function in MATLAB, respectively. We use the approximation of i.i.d MC with
$10^7$ samples as the pseudo ground-truth. The approximation errors $|\widehat{Z}-Z|/Z$ are shown in Fig.(e)-(h). our method
consistently outperforms MC, HMC and SS on all cases. Moreover, our method is much cheaper than SS and HMC.

**Q4. (R4) Comparison to sequential Monte Carlo.** When the positive density region takes a large fraction of the
entire domain, our method is very competitive (see Q3). When it is only inside a small part of a large domain, our
method may not be better than sequential adaptive sampling. In this case, it is interesting to take advantage of both
lattice and adaptive sampling. E.g., one can employ our subgroup rank-1 lattice as a rough partition of the domain to
find high mass regions, then take sequential adaptive sampling on the promising regions with the lattice points as the
start points. Also, it is interesting to consider progressively apply our subgroup rank-1 lattice to refine the partition.

**Q5. (All) Benefits to NeurIPS community.** Our subgroup rank-1 lattice performs good and robust. It does not
have any hyperparameter and is very convenient and cheap for points set construction. It has potential applications
at Bayesian inference, kernel approximation and the approximation of Wasserstein distance. It may also be able to
combine with sequential MC as discussed in Q4. Readers may be inspired by or learned from our technique.

[Meta-Review · NeurIPS 2020]

This paper is very much borderline and sparked an extensive discussion among reviewers. On the positive side, this work presents a simple closed form generation rule for rank-1 lattice in QMC, which previously required exhaustive search. The method is novel and solid, with promising empirical results. On the negative sides, all reviewers have concerns with 1) the lack of comparison to methods in ML community, and the fitness of the venue (however, the target problem of integration approximation is of high importance in Neurips as well); 2) some limited clarity/questions on empirical methodology; and 3) some writing quality /typo issues. The authors did an excellent job in rebuttal, including providing some initial results on a toy restricted Boltzmann model. We encourage the authors to add more comparisons to MC/MCMC methods that are more widely used in machine learning and discuss its applicability to the high dimensional and graph-structured integration problems in machine learning. We hope this work can provide an opportunity for Neurips audience to learn about QMC.